# B and T Cell Bi-Cistronic Multiepitopic Vaccine Induces Broad Immunogenicity and Provides Protection Against SARS-CoV-2

**DOI:** 10.3390/vaccines12111213

**Published:** 2024-10-25

**Authors:** Beatriz Perdiguero, Enrique Álvarez, Laura Marcos-Villar, Laura Sin, María López-Bravo, José Ramón Valverde, Carlos Óscar S. Sorzano, Michela Falqui, Rocío Coloma, Mariano Esteban, Susana Guerra, Carmen Elena Gómez

**Affiliations:** 1Department of Molecular and Cellular Biology, Centro Nacional de Biotecnología (CNB), Consejo Superior de Investigaciones Científicas (CSIC), 28049 Madrid, Spain; enrique.alvarez@cnb.csic.es (E.Á.); lmarcos@cnb.csic.es (L.M.-V.); mesteban@cnb.csic.es (M.E.); 2Centro de Investigación Biomédica en Red de Enfermedades Infecciosas (CIBERINFEC), Instituto de Salud Carlos III (ISCIII), 28029 Madrid, Spain; laura.sin@ciberinfec.es; 3Department of Microbial Biotechnology, CNB-CSIC, 28049 Madrid, Spain; mlbravo@cnb.csic.es; 4Scientific Computing, CNB-CSIC, 28049 Madrid, Spain; jrvalverde@cnb.csic.es; 5Biocomputing Unit and Computational Genomics, CNB-CSIC, 28049 Madrid, Spain; coss@cnb.csic.es; 6Department of Preventive Medicine, Public Health and Microbiology, Faculty of Medicine, Universidad Autónoma de Madrid, 28029 Madrid, Spain; michelafalqui@gmail.com (M.F.); rocio.coloma@uam.es (R.C.); susana.guerra@uam.es (S.G.); 7Department of Microbiology, Icahn School of Medicine at Mount Sinai, New York, NY 10029, USA; 8Global Health and Emerging Pathogens Institute, Icahn School of Medicine at Mount Sinai, New York, NY 10029, USA

**Keywords:** SARS-CoV-2, vaccine, multiepitopic protein, DNA and MVA vectors, innate response, cellular response, immunogenicity, efficacy in mice, ISG15 adjuvant

## Abstract

Background: The COVID-19 pandemic, caused by SARS-CoV-2, has highlighted the need for vaccines targeting both neutralizing antibodies (NAbs) and long-lasting cross-reactive T cells covering multiple viral proteins to provide broad and durable protection against emerging variants. Methods: To address this, here we developed two vaccine candidates, namely (i) DNA-CoV2-TMEP, expressing the multiepitopic CoV2-TMEP protein containing immunodominant and conserved T cell regions from SARS-CoV-2 structural proteins, and (ii) MVA-CoV2-B2AT, encoding a bi-cistronic multiepitopic construct that combines conserved B and T cell overlapping regions from SARS-CoV-2 structural proteins. Results: Both candidates were assessed in vitro and in vivo demonstrating their ability to induce robust immune responses. In C57BL/6 mice, DNA-CoV2-TMEP enhanced the recruitment of innate immune cells and stimulated SARS-CoV-2-specific polyfunctional T cells targeting multiple viral proteins. MVA-CoV2-B2AT elicited NAbs against various SARS-CoV-2 variants of concern (VoCs) and reduced viral replication and viral yields against the Beta variant in susceptible K18-hACE2 mice. The combination of MVA-CoV2-B2AT with a mutated ISG15 form as an adjuvant further increased the magnitude, breadth and polyfunctional profile of the response. Conclusion: These findings underscore the potential of these multiepitopic proteins when expressed from DNA or MVA vectors to provide protection against SARS-CoV-2 and its variants, supporting their further development as next-generation COVID-19 vaccines.

## 1. Introduction

The coronavirus disease 2019 (COVID-19) pandemic, caused by the severe acute respiratory syndrome coronavirus-2 (SARS-CoV-2), has led to significant global health, economic and social impacts. As of 6 October 2024, over 776 million cases and more than 7 million deaths have been reported worldwide (https://COVID19.who.int; accessed on 21 October 2024). While first-generation vaccines targeting the SARS-CoV-2 spike (S) protein, such as mRNA-1273, BNT162b2, Ad26.CoV2 and NVX-CoV2373, have been instrumental in reducing disease severity and mortality, they face limitations. These include diminished efficacy against emerging variants, waning antibody levels and the need for frequent updates to target new VoCs [1].

Evidence suggests that immune responses to SARS-CoV-2 in previously infected individuals are more durable and cross-reactive than those induced by S protein-based vaccines. Recovered COVID-19 patients mount a broad and durable immunity after infection, characterized by NAbs with an extended half-life of over 200 days, indicating the presence of long-lived plasma cells that contribute to sustained antibody levels. Additionally, SARS-CoV-2-specific polyfunctional CD4 and CD8 T cells targeting multiple regions of the S protein and other viral proteins, such as membrane (M), nucleocapsid (N) and ORF3a proteins, are maintained over time, leading to a more diverse antibody and T cell repertoire [2,3]. This underscores the importance of developing vaccine strategies that not only induce NAbs but also robust and cross-reactive T cell responses, targeting multiple viral antigens to ensure comprehensive and long-lasting protection against SARS-CoV-2 and its variants.

Epitope-focused vaccines represent a cutting-edge approach in vaccine development, aiming to narrow the immune response to critical neutralizing and T cell epitopes. By specifically targeting conserved and functional regions within the viral proteome, these vaccines address the challenge of viral diversity and enhance the breadth and efficacy of the immune response, minimizing the risks of antigenic drift and immune evasion, which are common challenges with traditional vaccine strategies. Furthermore, by focusing on key B and T cell epitopes from different antigens, these polyvalent multiepitopic vaccines can induce a more precise and robust activation of cross-reactive humoral and cellular immunity, offering protection against diverse and evolving viral variants. This approach also ensures a good safety profile by avoiding the generation of non-protective or harmful responses [4,5,6].

In a previous study, we aimed to broaden the capacity of vaccines based on the single S protein by designing CoV2-BMEP, a multiepitopic synthetic construct containing dominant and persistent B cell epitopes from conserved regions of SARS-CoV-2 S, M and N structural proteins, that was expressed from DNA or MVA (Modified Vaccinia virus Ankara) vectors for preclinical evaluation [7]. In C57BL/6 mice, both homologous and heterologous prime/boost vector combinations induced SARS-CoV-2-specific CD4 and CD8 T cells, with MVA-CoV2-BMEP/MVA-CoV2-BMEP regimen eliciting the highest SARS-CoV-2-specific CD8 T cell responses in spleen and detectable binding antibodies (BAbs) against SARS-CoV-2 S and N proteins. In susceptible K18-hACE2 transgenic (Tg) mice, two doses of MVA-CoV2-BMEP induced S- and N-specific BAbs and NAbs against different VoCs. After SARS-CoV-2 challenge, vaccinated animals with high titers of NAbs were fully protected against mortality, correlating with the inhibition of the cytokine storm and a reduction in virus infection in the lungs [7].

As a follow-up to our previous work, we describe here the design and preclinical evaluation of two novel polyvalent multiepitopic proteins, CoV2-TMEP and CoV2-BMEP-P2A-TMEP. The CoV2-TMEP candidate, expressed from a DNA vector (DNA-CoV2-TMEP), includes immunodominant T cell regions containing overlapping conserved CD4 T helper and CD8 CTL epitopes from the SARS-CoV-2 S, M, N and envelope (E) structural proteins, which are recognized across a wide range of HLA class I and II alleles. This design ensures high global population coverage and cross-protection against diverse coronavirus strains. The bi-cistronic CoV2-BMEP-P2A-TMEP candidate, expressed from the MVA vector (MVA-CoV2-B2AT), combines overlapping B and T cell regions in a single immunogen to target both arms of the immune response. It produces a single fusion protein containing both the CoV2-BMEP, previously described in [7], and CoV2-TMEP constructs, linked by a P2A sequence. The P2A sequence facilitates a “self-cleaving” process during translation, resulting in the production of the two proteins independently from a single transcript. Both candidates were evaluated in vitro in cultured cells and in vivo in a mouse model.

In addition, we have previously demonstrated that the ubiquitin-like molecule ISG15 functions as an immune adjuvant in the context of HIV immunization protocols, enhancing antigen-specific CD8 T cell responses [8]. In the present study, we have also demonstrated that the co-expression of a mutated form of ISG15, which is unable to be conjugated, and CoV2-BMEP-P2A-TMEP construct leads to an enhanced immune response to SARS-CoV-2 vaccines, underscoring the critical importance of selecting appropriate adjuvants for optimizing immunization protocols.

## 2. Materials and Methods

### 2.1. Design of Chimeric CoV2-TMEP Polyvalent Multiepitopic Immunogen Targeting T Cells

The selection of viral fragments for the CoV2-TMEP multiepitopic vaccine candidate was based on the following criteria: (i) the inclusion of major persistent and immunodominant antigenic regions within the SARS-CoV-2 proteome that have been associated with survival in patients infected with SARS-CoV-1 and SARS-CoV-2, with high conservation across human endemic coronaviruses (HCoVs) and low mutation rates; (ii) the presence of CD4 and CD8 T cell epitopes, restricted by a wide range of HLA class I and II molecules, to induce cross-protective immune responses and ensure broad global population coverage; and (iii) the exclusion of decoy epitopes that are irrelevant for viral control and could compromise vaccine safety. Thus, for the design of the CoV2-TMEP vaccine candidate, we selected major persistent and immunodominant T cell regions overlapping conserved CD4 T helper and CD8 cytotoxic T lymphocyte (CTL) epitopes that bind multiple HLA class I and class II alleles, derived from SARS-CoV-2 structural proteins S, M, N and E. These epitopes were consistently recognized by SARS-CoV-1 and SARS-CoV-2 survivors during natural infection or following vaccination and were identified through the manual sequence inspection of several research articles previously referenced by Perdiguero et al. [7].

The S, M, N and E protein sequences of seven known HCoV strains, including HCoV-HKU1 (NC_002645), HCoV-NL63 (NC_005831), HCoV-OC43 (NC_006213), HCoV-229E (NC_002645), MERS-CoV (NC_019843), SARS-CoV (NC_0041718) and SARS-CoV-2 (NC_045512) were obtained from the National Center for Biotechnology Information (NCBI). Sequence alignment was performed using ClustalW (version 2.0.12) with default settings to identify major similarity blocks. Common epitopes that remained reactive in over 80% of COVID-19 samples up to 180–220 days post-symptom onset were defined as dominant and persistent by Li et al. [9]. CoV2-TMEP includes 12 regions from structural proteins: 3 from the S protein (S2 domain), 2 from the E protein, 3 from the M protein and 4 from the N protein (Table 1), connected by dual or triple alanine (A) residues as cleavable linkers to optimize cleavage sites, enhance processing of the fusion protein, and reduce altered bioactivity due to epitope juxtaposition. A mutant form of the signal peptide (SP) of human tissue plasminogen activator (tPA-22P/A SP) was added at the N-terminus, as it has been shown to significantly enhance the secretion of heterologous antigens [10], and a FLAG-tag was included at the C-terminus for expression analysis (Figure 1A).

### 2.2. In Silico Characterization of CoV2-TMEP Synthetic Protein

The in silico characterization of the CoV2-TMEP multiepitopic protein has been performed as previously described [7]. To predict the physicochemical properties of the CoV2-TMEP vaccine candidate, the ExPASy ProtParam online tool (https://web.expasy.org/protparam/; accessed on 5 February 2023) was employed. Parameters analyzed included the average molecular weight (MW), theoretical isoelectric point (pI), amino acid residue count, extinction coefficients, aliphatic index, instability index (II), grand average of hydropathicity (GRAVY) and estimated in vitro and in vivo half-life. The antigenicity of CoV2-TMEP was predicted using the ANTIGENpro bioinformatics tool (http://scratch.proteomics.ics.uci.edu/; accessed on 5 February 2023), which utilizes a pathogen-independent, sequence-based, alignment-free approach [11]. AllergenFP server (http://www.ddgpharmfac.net/AllergenFP/; accessed on 5 February 2023) was used for allergenicity prediction [12].

The CoV2-TMEP sequence was further analyzed to confirm the prediction of antigenic sites using the imed.med.ucm.es server (http://imed.med.ucm.es; accessed on 16 February 2023), along with epitope prediction and Immune Epitope Database (IEDB) analysis tools from immuneepitope.org [13] and tepitool [14] for human MHC classes I and II (DP, DQ and DR) binding predictions. Additionally, a predicted worldwide population coverage analysis was performed using data from the IEDB Database (as of 11 August 2020 with the recommended IEDB methods. The predicted MHC class I and II epitope coverage for the CoV2-TMEP construct is listed in Appendix A.

### 2.3. Design and In Silico Characterization of the Bi-Cistronic CoV2-BMEP-P2A-TMEP Synthetic Protein

The bi-cistronic CoV2-BMEP-P2A-TMEP construct includes *CoV2-BMEP* and *CoV2-TMEP* genes linked by a P2A linker sequence (GSGATNFSLLKQAGDVEENPGP) [15]. CoV2-BMEP protein includes 11 antigenic regions: 8 from the S protein (2 from S1 and 6 from S2 domains), 1 from the M protein and 2 from the N protein connected by the flexible GSGSG linker and ordered according to the SARS-CoV-2 genomic organization [7]. tPA-22P/A SP was added at the N-terminus and HA-tag at the C-terminus. For the inclusion of CoV2-BMEP in the CoV2-BMEP-P2A-TMEP construct, the stop codon and the HA-tag were removed. CoV2-TMEP sequence in the CoV2-BMEP-P2A-TMEP construct was modified as follows: (i) the region corresponding to the putative SARS-CoV-2 viroporin encoded by *ORF-E* was excluded (E_13–29_); (ii) the region E_38–59_ was extended to E_38–65_; (iii) the FLAG-tag at the N-terminus was replaced by the HA-tag; and (iv) tPA-22P/A SP was replaced by the synthetic Secrecon SP [16] to avoid potential recombination events between SP from both CoV2-BMEP and CoV2-TMEP constructs. The in silico characterization of CoV2-BMEP-P2A-TMEP synthetic protein was performed as described above for CoV2-TMEP construct.

### 2.4. Cells

The following cell lines were used in this study: HEK-293T (ATCC catalog no. CRL-3216), HeLa (ATCC CCL-2), DF-1 (ATCC CRL-12203) and Vero-E6 (ATCC C1008; Vero 76, clone E6). For the culture of these cells, Dulbecco’s Modified Eagle’s Medium (DMEM; Sigma-Aldrich, St. Louis, MO, USA) was supplemented with 100 U/mL of penicillin/100 µg/mL of streptomycin (Sigma-Aldrich), 0.5 μg/mL of amphotericin B (Fungizone; Gibco-Life Technologies, Waltham, MA, USA), 0.1 mM of non-essential amino acids (Sigma-Aldrich), 2 mM of L-glutamine (Sigma-Aldrich) and 10% heat-inactivated fetal bovine serum (FBS; Gibco-Life Technologies) for HEK-293T, DF-1 and Vero-E6 cells or 10% heat-inactivated newborn calf serum (NCS; Sigma-Aldrich) for HeLa cells. All cell lines were maintained in a controlled environment at 37°C, 5% CO_2_ and humidified atmosphere.

### 2.5. Viruses

The following poxviruses were used in this work: the attenuated wild-type MVA (MVA-WT), which derives from the Ankara strain after 586 serial passages in chicken embryo fibroblasts (provided by Prof. Dr. Gerd Sutter, Ludwig-Maximilians-University of Munich, Munich, Germany); the recombinant virus MVA-CoV2-BMEP, which contains the *CoV2-BMEP* gene inserted into the thymidine kinase (TK) locus (*J2R* gene) of the parental MVA-WT, as previously described [7]; the recombinant MVA-Δ3-GFP (kindly provided by Dr. Juan García-Arriaza, CNB-CSIC, Madrid, Spain), which has deletions in the vaccinia virus (VACV) immunomodulatory genes *C6L*, *K7R* and *A46R*, along with the green fluorescent protein (*GFP*) gene inserted into the TK locus of the MVA-WT genome [17]; the recombinant viruses MVA-Δ3-ISG15GG and MVA-Δ3-ISG15AA (kindly provided by Dr. Susana Guerra, UAM, Madrid, Spain) with murine *Isg15gg* (wild-type ISG15 form; able to perform ISGylation) or *Isg15aa* (mutated ISG15 form; unable to perform ISGylation) genes inserted in the TK locus of MVA-Δ3-GFP genome, as previously described [8]; and the recombinant MVA-CoV2-BMEP-P2A-TMEP (referred to as MVA-CoV2-B2AT), in which the *CoV2-BMEP-P2A-TMEP* gene was inserted into the TK locus of the MVA-WT genome, and generated in this study. Poxviruses were propagated in DF-1 cells with supplemented DMEM containing 2% FBS.

The SARS-CoV-2 strain MAD6 (kindly provided by Prof. Luis Enjuanes and Dr. José M. Honrubia, CNB-CSIC, Madrid, Spain) was isolated from a nasopharyngeal swab of a 69-year-old male COVID-19 patient from Hospital 12 de Octubre (Madrid, Spain), as previously reported [18]. The SARS-CoV-2 Beta (B.1.351) variant (hCoV-19/France/PDL-IPP01065i/2021, 10H/501Y.V2) was supplied by the National Reference Centre for Respiratory Viruses at the Institut Pasteur (Paris, France), headed by Pr. Sylvie van der Werf, and Dr. Jérôme Besson from the Bioliance, st-Herblain Laboratory (Saint-Herblain, France) provided the human sample from which the virus was isolated. The SARS-CoV-2 VoCs Delta (B.1.617) (SARS-CoV-2, Human, 2021, Germany ex India, 20A/452R) and Omicron BA.1 (B.1.1.529) (hCoV-19/Belgium/rega-20174/2021, EPI_ISL_6794907) were kindly provided by Dr. Juan García-Arriaza (CNB-CSIC, Madrid, Spain), as previously reported [19].

### 2.6. DNA Vectors

The CoV2-TMEP and CoV2-BMEP-P2A-TMEP sequences were codon-optimized for expression in human or VACV, respectively. They were chemically synthesized and subcloned into the pcDNA vector by GeneArt (Thermo Fisher Scientific, Waltham, MA, USA), resulting in the plasmids pcDNA-CoV2-TMEP (referred to as DNA-CoV2-TMEP) and pcDNA-CoV2-BMEP-P2A-TMEP. For the generation of the pLeoLZ-BMEP-P2A-TMEP plasmid transfer vector, the synthetic *CoV2-BMEP-P2A-TMEP* gene was excised from plasmid pcDNA-CoV2-BMEP-P2A-TMEP using BamHI and NotI restriction endonucleases and inserted into the pLeoLZ vector (previously digested with BglII and NotI; [20]). The resulting plasmid pLeoLZ-BMEP-P2A-TMEP was checked by PCR and DNA sequence analysis (Macrogen, Seoul, Republic of Korea). It directs the insertion of *CoV2-BMEP-P2A-TMEP* gene into the viral TK locus of MVA-WT genome. The pcDNA3.0 vector (Invitrogen, Carlsbad, CA, USA; referred to as DNA-ϕ) was used as control. Plasmids were purified (EndoFree Plasmid Mega kit; Qiagen, Hilden, Germany) and diluted for injection in endotoxin-free phosphate-buffered saline (PBS).

### 2.7. Construction and In Vitro Characterization of MVA-CoV2-B2AT Recombinant Virus

The construction of MVA-CoV2-B2AT virus was achieved by homologous recombination, as previously described [21]. In brief, 1 × 10^6^ DF-1 cells were infected with MVA-WT at 0.01 plaque-forming units (PFUs) per cell and transfected 1 h later with 4 µg of the plasmid pLeoLZ-BMEP-P2A-TMEP described above using Lipofectamine-2000 (Invitrogen) according to the manufacturer’s instructions. At 72 h post-infection (hpi), cells were collected, lysed by freeze–thaw cycling, sonicated and then used for screening MVA recombinants. During the first three plaque purification steps, MVA recombinant viruses that transiently co-expressed β-galactosidase (β-Gal; *lacZ* marker gene) and contained the *CoV2-BMEP-P2A-TMEP* gene were isolated from DF-1 cells stained with 5-bromo-4-chloro-3-indolyl β-D-galactopyranoside (X-Gal; 1.2 mg/mL; Sigma-Aldrich). In the subsequent three isolation steps, MVA recombinants with a deleted *lacZ* gene and containing *CoV2-BMEP-P2A-TMEP* gene were identified by screening for non-stained viral plaques in the presence of X-Gal. The resulting MVA-CoV2-B2AT virus was cultured in DF-1 cells and the viral crude preparation obtained (P2 stock) was used to expand the virus in large-scale cultures of DF-1 cells, followed by purification through two 36% (*w*/*v*) sucrose cushions (P3 stock). Viral titers were determined using an immunostaining plaque assay, as previously described [22]. The viral stocks were confirmed to be free of contamination from mycoplasma, bacteria and fungi.

To verify the identity and purity of the MVA-CoV2-B2AT viral preparation, DNA was extracted from DF-1 cells infected with either MVA-WT or MVA-CoV2-B2AT (P2 and P3 stocks) at 5 PFUs per cell for 24 h, as previously described [23]. Viral TK locus was analyzed by PCR using the extracted DNA as template. The amplification reactions were performed using Phusion High-Fidelity DNA polymerase (BioLabs, Ipswich, MA, USA), according to the manufacturer’s instructions, using the primers TK-L: 5′-TGATTAGTTTGATGCGATTC-3′ and TK-R: 5′-CTGCCGTATCAAGGACA-3′, which span the TK flanking regions.

The stability of the CoV2-BMEP-P2A-TMEP protein expressed by the MVA-CoV2-B2AT virus was assessed after serial infection passages at a low multiplicity of infection (MOI) in DF-1 cells cultured in T25 tissue culture flasks, as previously described [24]. The correct expression of the CoV2-BMEP-P2A-TMEP protein in 24 isolated plaques from stability passage 7 was evaluated by Western blotting analysis using a rabbit polyclonal antibody against the SARS-CoV-2 N protein (1:1000; Sino Biological, Beijing, China) for CoV2-BMEP detection or a rabbit polyclonal anti-HA antibody (1:1000; Sigma-Aldrich) for the detection of CoV2-TMEP, followed by a goat anti-rabbit-Horseradish peroxidase (HRP) antibody (1:5000; Sigma-Aldrich).

### 2.8. Expression Analysis of CoV2-TMEP and CoV2-BMEP-P2A-TMEP Multiepitopic Proteins by Western Blotting

To determine the correct expression of the CoV2-TMEP protein, HEK-293T cells were transfected with 1 μg of either DNA-ϕ or DNA-CoV2-TMEP using Lipofectamine-2000 (Invitrogen) according to the manufacturer’s instructions. Cells were collected at 24, 48 and 72 h post-transfection (hpt), and both cellular pellets and supernatants were processed as previously described [25], fractionated by 12% Sodium Dodecyl Sulfate-Polyacrylamide Gel Electrophoresis (SDS-PAGE) and analyzed by Western blotting using a mouse monoclonal anti-FLAG M2 antibody (1:1000; Sigma-Aldrich), followed by an HRP-conjugated anti-mouse antibody (1:2000; Sigma-Aldrich) to assess CoV2-TMEP expression. The immunocomplexes were detected using an enhanced-chemiluminescence system (ECL Plus; GE Healthcare, Chicago, IL, USA).

To evaluate the potential association between CoV2-TMEP and NLRP3, as well as the induction of PARP cleavage, monolayers of HEK-293T were transfected as described above, harvested at 24 hpt and analyzed by Western blotting using a rabbit polyclonal anti-NLRP3/NALP3 antibody (1:200; Cat. #NBP2-12446; Novus Biologicals, Centennial, CO, USA) or a rabbit polyclonal anti-PARP antibody (1:500; Cat. #9542; Cell Signaling, Danvers, MA, USA), followed by an HRP-conjugated anti-rabbit antibody (1:5000; Sigma-Aldrich).

To assess the correct expression of CoV2-BMEP-P2A-TMEP protein, DF-1 cells were infected with MVA-CoV2-B2AT at 3 PFUs per cell. Cellular extracts were collected at 0, 6, 16 and 24 hpi and analyzed by Western blotting using a rabbit polyclonal anti-SARS-CoV-2 N antibody (1:1000; Sino Biological) for CoV2-BMEP detection or a rabbit polyclonal anti-HA antibody (1:1000; Sigma-Aldrich) for the detection of CoV2-TMEP, followed by an HRP-conjugated anti-rabbit antibody (1:5000; Sigma-Aldrich).

### 2.9. Subcellular Localization of CoV2-TMEP and CoV2-BMEP-P2A-TMEP Multiepitopic Proteins by Immunofluorescence Microscopy

To determine the subcellular localization of CoV2-TMEP protein, HeLa cells grown on glass coverslips were transfected with 0.5 μg of either DNA-ϕ or DNA-CoV2-TMEP using Lipofectamine-2000 (Invitrogen), following the manufacturer’s instructions. Cells were harvested at 24, 48 and 72 hpt, and then processed for immunofluorescence microscopy as previously reported [7]. Briefly, cells were fixed with 4% paraformaldehyde (PFA), permeabilized with PBS 1X-0.25% saponin and incubated with a mouse monoclonal anti-FLAG M2 antibody (1:200; Sigma-Aldrich), followed by incubation with a goat anti-mouse Alexa Fluor 488 secondary antibody (diluted 1:500; Life Technologies, Carlsbad, CA, USA; green fluorescence staining). Cell nuclei were then stained with 4′,6-diamidino-2-phenylindole (DAPI; 1:200; Sigma-Aldrich), and coverslips were finally mounted onto glass slides using ProLong Gold anti-fade reagent (Invitrogen). The optical sections of the cells were acquired with a Leica TCS SP5 microscope, and LAS AF v4.7.0.28176 software (Leica Microsystems, Wetzlar, Germany) was used for image recording and processing. 

To explore the potential association between CoV2-TMEP and NLRP3, HeLa cells were transfected as described above, harvested at 24 hpt and processed for confocal microscopy using a mouse monoclonal anti-FLAG M2 antibody (Sigma-Aldrich) and a rabbit polyclonal anti-NLRP3/NALP3 antibody (1:50; Novus Biologicals). FLAG was detected with a goat anti-mouse Alexa Fluor 488 secondary antibody (diluted 1:500; Life Technologies; green fluorescence staining), while NLRP3/NALP3 was detected using a goat anti-rabbit Alexa Fluor 594 secondary antibody (diluted 1:500; Invitrogen; red fluorescence staining). Image colocalization analysis was achieved using LAS X v4.7.0.28176 software with the colocalization license, and ImageJ/Fiji software was used to calculate Pearson’s correlation coefficient for colocalization.

To analyze the subcellular localization of CoV2-BMEP-P2A-TMEP protein, HeLa cells were grown on glass coverslips and infected with MVA-CoV2-B2AT at 1 PFU per cell. Cells were harvested at 16 hpi and stained with wheat germ agglutinin (WGA) conjugated to Alexa Fluor 555 (diluted 1:500; Invitrogen; red fluorescence staining) for 2 min at room temperature (RT). Cells were then labeled with either a rabbit polyclonal antibody against SARS-CoV-2 N protein (1:200; Sino Biological) or a rabbit polyclonal anti-HA antibody (1:200; Sigma-Aldrich), followed by a goat anti-rabbit Alexa Fluor 488 secondary antibody (diluted 1:500; Life Technologies; green fluorescence staining). Finally, DAPI (1:200; Sigma-Aldrich) was used to stain cell nuclei.

### 2.10. Peptides

The SARS-CoV-2 peptide pools used for in vivo studies include S1 (158 peptides), S2 (157 peptides), M (53 peptides), N (102 peptides) and E (16 peptides) (JPT Peptide Technologies GmbH, Berlin, Germany). They covered the SARS-CoV-2 S, M, N and E proteins incorporated into the CoV2-TMEP and CoV2-BMEP proteins as consecutive 15-mer peptides, overlapping by 11 amino acids.

### 2.11. Ethics Statement

The immunogenicity and efficacy mouse studies using C57BL/6JOlaHsd and K18-hACE2 Tg mice, respectively, were approved by the Ethical Committee of Animal Experimentation (CEEA) at CNB-CSIC and at Centro de Investigación en Sanidad Animal (CISA)-Instituto Nacional de Investigación y Tecnología Agraria y Alimentaria (INIA-CSIC) (Valdeolmos, Madrid, Spain) and by the Division of Animal Protection of the Comunidad de Madrid with permit numbers PROEX 169.4/20 and 161.5/20. All procedures adhered to EU guidelines and Spanish law under Royal Decree RD 53/2013.

### 2.12. Preclinical Evaluation of CoV2-TMEP and CoV2-BMEP-P2A-TMEP Proteins in Mice

#### 2.12.1. Analysis of the Impact of CoV2-TMEP or CoV2-BMEP-P2A-TMEP Expression on Immune Cell Recruitment by Flow Cytometry

To assess the impact of CoV2-TMEP expression on immune cell recruitment, a group of female C57BL/6JOlaHsd mice (6–8 weeks old; n  =  9), purchased from Envigo Laboratories (Sant Feliu de Codines, Barcelona, Spain), were immunized intramuscularly (i.m.) with 25 μg of DNA-CoV2-TMEP (group 1). PBS-treated mice were used as the control group (group 2). On days 1, 3 and 7 post-inoculation, 3 mice from each group were sacrificed and the muscle tissue from the injection site and the inguinal draining lymph nodes (DLNs) were harvested and processed for flow cytometry analysis. To determine the effect of CoV2-BMEP-P2A-TMEP expression on immune cell recruitment, two groups of female C57BL/6JOlaHsd mice (6–8 weeks old; n  =  4) were immunized i.m. with 1 × 10^7^ PFUs of either MVA-WT or MVA-CoV2-B2AT (groups 1 and 2). A PBS-treated control group was included (group 3). On day 1 post-inoculation, the mice were sacrificed, and the muscle tissue and inguinal DLNs were collected and processed. Muscle samples were mechanically dissociated followed by enzymatic digestion as previously described [7], and DLNs were meshed through a cell strainer.

Flow cytometry was performed to analyze the different immune cell populations recruited to the muscle and DLNs after immunization with DNA-CoV2-TMEP or MVA-CoV2-B2AT, as previously described [7]. Briefly, 1 × 10^6^ cells per well were seeded on 96-well plates and incubated with the LIVE/DEAD Fixable Red Dead Cell Stain Kit (Invitrogen) at 4 °C for 30 min. After blocking Fc receptors using anti-CD16/CD32 antibody (BD Biosciences, San Jose, CA, USA), cells were incubated with MHC-II-biotin at 4 °C for 20 min, followed by incubation with specific fluorochrome-conjugated surface antibodies for 15 min at 4 °C for the identification of various myeloid immune cell populations (BCs, TCs, NKs, moCs, DCs and NOs), including: Ly6G-PE, CD19-PE, CD3-PE, SinglecF-PE, Ly6C-PerCP, avidin-PE-Cy7, CD64-APC, CD11b-Alexa Fluor 700, CD11c-APC-Cy7, CD45-PB and B220-BV510. All antibodies were sourced from BD Biosciences. Finally, the samples were acquired using a Gallios flow cytometer (Beckman Coulter, Brea, CA, USA), and data were analyzed using FlowJo software (version 10.4.2; Tree Star, Ashland, OR, USA). Cell-gated events ranged between 10^5^ and 5 × 10^5^. The different immune cell populations were identified following the gating strategies previously described [7].

#### 2.12.2. SARS-CoV-2-Specific Immunogenicity Elicited by CoV2-TMEP or CoV2-BMEP-P2A-TMEP Proteins

To characterize the SARS-CoV-2-specific immunogenicity elicited by the CoV2-TMEP protein, two groups of female C57BL/6JOlaHsd mice (6–8 weeks old; n  =  3) were immunized i.m. with 50 μg of either DNA-CoV2-TMEP or DNA-ϕ on days 0 and 115. Twelve days post-boost (day 127), the animals were sacrificed, and spleens and lungs were processed for intracellular cytokine staining (ICS) assay. For the analysis of the SARS-CoV-2-specific immunogenicity induced by the CoV2-BMEP-P2A-TMEP protein, four groups of female C57BL/6JOlaHsd mice (6–8 weeks old; n  =  5) were immunized i.m. with 1×10^7^ total PFUs of the following MVA-based vector combinations: G1: MVA-CoV2-B2AT + MVA-Δ3-GFP; G2: MVA-CoV2-B2AT + MVA-Δ3-ISG15GG; G3: MVA-CoV2-B2AT + MVA-Δ3-ISG15AA and G4: MVA-Δ3-GFP. This was carried out on days 0 and 28. Twelve days post-boost (day 40), the animals were sacrificed and spleens were processed for ICS assay. Lung samples were mechanically dissociated followed by enzymatic digestion as previously reported [7], while spleens were processed by meshing through a cell strainer.

The analysis of the magnitude and phenotype of the SARS-CoV-2-specific T cell acute immune responses by flow cytometry has been previously described in [7]. In brief, 2 × 10^6^ splenocytes or lung-derived lymphocytes (erythrocyte-depleted) per well were seeded on 96-well plates and stimulated ex vivo for 6 h in complete RPMI 1640 medium (Sigma-Aldrich) containing 10% FBS, 1 µL/mL of Golgiplug (BD Biosciences), anti-CD107a-FITC (BD Biosciences), monensin 1X (Invitrogen) and 1 µg/mL of the SARS-CoV-2 peptide pools representing the S, M, N and E antigens (JPT Peptide Technologies GmbH). After stimulation, lymphocytes were incubated with surface markers, fixed/permeabilized (Cytofix/Cytoperm kit; BD Biosciences) and intracellularly stained. The fluorochrome-conjugated antibodies CD3-PE-CF594, CD4-APC-Cy7 and CD8-V500 were used for phenotypic analyses, and IFN-γ-PeCy7, IL-2-APC and TNF-α-PE were used for functional analyses (all from BD Biosciences). Dead cells were excluded using the LIVE/DEAD Fixable Violet Dead Cell Stain kit (Invitrogen). Cells were acquired using a GALLIOS flow cytometer (Beckman Coulter), and data were analyzed using FlowJo software (Version 10.4.2; Tree Star). Background responses from unstimulated controls (RPMI) were subtracted from responses in stimulated samples. To analyze the polyfunctional profile of the SARS-CoV-2-specific T cell responses, the Boolean combinations of single functional gates were generated to quantify the frequency of each response based on all possible combinations of cytokine production and CD107a expression. For each specific functional combination, background responses from the unstimulated controls (RPMI) were subtracted from those obtained in stimulated samples.

#### 2.12.3. Efficacy of MVA-CoV2-B2AT against SARS-CoV-2 Infection in Susceptible K18-hACE2 Tg Mice

For the efficacy study, three groups of female K18-hACE2 (034860-B6.Cg-Tg(K18-ACE2)2Prlman/J) Tg mice (6–8 weeks old; n = 6), purchased from Jackson Laboratory (Bar Harbor, ME, USA), were immunized i.m. with 1 × 10^7^ PFUs of either MVA-CoV2-BMEP (group 1) or MVA-CoV2-B2AT (group 2) or with PBS (group 3) on days 0 and 27. Blood samples were collected via submandibular bleeding at 26 days post-prime (d26) and 25 days post-boost (d52) to analyze SARS-CoV-2-specific NAbs. At 5 weeks post-boost (day 62), the mice were lightly isoflurane-anesthetized and challenged via intranasal (i.n.) route with 1 × 10^5^ PFUs of SARS-CoV-2 virus (Beta variant). Five days post-challenge, the mice were sacrificed, and lung, nasal turbinate and blood samples were collected and processed as previously described [7]. All experiments were performed using a laminar flow cabinet in the biosafety level 3 (BSL-3) facilities at CISA-INIA.

##### SARS-CoV-2 Neutralization Assay

Live virus SARS-CoV-2 NAbs in serum samples from individual K18-hACE2 Tg mice were assessed using a microneutralization test (MNT) assay performed in a BSL-3 laboratory as previously described [26]. Briefly, samples were 2-fold serially diluted in duplicates and incubated with 100 TCID50 of SARS-CoV-2 MAD6 or Beta (B.1.351), Delta (B.1.617) or Omicron BA.1 variants for 1 h at 37 °C. After this, mixtures were added to Vero-E6 cells and incubated for 3 days at 37 °C and 5% CO_2_. Next, cells were fixed with 10% formaldehyde (Sigma-Aldrich) for 1 h, stained using crystal violet (Sigma-Aldrich) and, after drying the plates, H_2_O-10% SDS was used to dilute the crystal violet. Finally, OD_570_ was measured using a luminometer. To determine neutralization titers, the half maximal inhibitory concentration (IC_50_) and 95% confidence intervals (95% CI) were determined using a nonlinear regression model fit with settings for a log agonist versus normalized response curve with GraphPad Prism v10.1.0 Software (GraphPad Software, San Diego, CA, USA).

##### Analysis of SARS-CoV-2 RNA by Reverse Transcription-Quantitative Polymerase Chain Reaction (RT-qPCR)

The analysis of SARS-CoV-2 RNA by RT-qPCR was performed as previously described [7]. Briefly, lung tissues from K18-hACE2 Tg mice challenged with SARS-CoV-2 were collected in RNALater (Sigma-Aldrich) and homogenized with a gentleMACS dissociator (Miltenyi Biotec, Bergisch Gladbach, Germany) using RLT buffer (Qiagen) containing β-mercaptoethanol (Sigma-Aldrich). The samples were then processed for RNA extraction as previously described [27]. The presence of SARS-CoV-2 viral RNA was assessed using validated probes and specific set of primers targeting the genomic RNA dependent RNA polymerase gene (*RdRp*) or the subgenomic E protein gene (*E*) of SARS-CoV-2. The cellular 28S rRNA was used as a normalization control [28]. Data collection was carried out with a 7500 real-time PCR system (Applied Biosystems, Waltham, MA, USA) and analysis was performed using the 7500 software v2.0.6. Relative RNA arbitrary units (A.U.) were calculated relative to a naïve lung sample using the 2^−ΔΔCt^ method. Each sample was evaluated in duplicate.

##### Analysis of SARS-CoV-2 Virus Yields by Plaque Assay

The presence of SARS-CoV-2 infectious virus in lung and nasal turbinate samples from K18-hACE2 Tg mice was analyzed using a plaque assay previously described [19]. Briefly, undiluted and serially 10-fold diluted homogenized lung or nasal turbinate tissue samples were added in triplicates to Vero-E6 cells. After 1 h, the inoculum was removed and the plates were incubated at 37 °C with 5% CO_2_ in 2:1 DMEM 2X-4% FBS: Avicel^®^ RC-591 (DuPont Nutrition Biosciences ApS, Kongens Lyngby, Denmark) for 4 days. Next, the cells were fixed with 10% formaldehyde (Sigma-Aldrich) and plaques were identified by adding 0.5% crystal violet (Sigma-Aldrich). SARS-CoV-2 titers were calculated as PFUs per milligram of lung tissue or per milliliter of nasal turbinate.

### 2.13. Data Analysis and Statistics

For the statistical analysis of flow cytometry data, a method that adjusts the values for the non-stimulated controls (RPMI) and calculates *p* values and confidence intervals was used [29]. Only SARS-CoV-2 antigen responses that were significantly higher than the corresponding RPMI values were represented and background-subtracted. NAbs titers and viral yield data were analyzed using an ordinary one-way ANOVA on transformed data, followed by Tukey’s multiple comparison test. RT-qPCR data were analyzed using an unpaired non-parametric *t* test with Welch’s correction. Graphs and statistical analyses were generated using GraphPad Prism v10.1.0 Software (GraphPad Software). Statistical significance is indicated as follows: *, *p* < 0.05; **, *p* < 0.005; and ***, *p* < 0.001.

## 3. Results

### 3.1. In Vitro Characterization of the Multiepitopic CoV2-TMEP Protein Expressed from a DNA Vector

To counteract SARS-CoV-2 infection and prolong the specific immune response, induction of both B and T cell responses with broad coverage is needed. For this reason, we have first designed and used different computational algorithms to characterize the polyvalent CoV2-TMEP multiepitopic protein as detailed in the Materials and Methods section. The CoV2-TMEP (Figure 1A and Table 1) linear sequence had a total length of 306 amino acids (~35 KDa) with a theoretical pI value of 9.50. The half-life was estimated to be 30 h in mammalian reticulocytes in vitro, and >20 h in yeast and >10 h in *E. coli* in vivo. An II of 36.42 was computed, classifying the protein as stable. The estimated aliphatic index was predicted to be 102.22, indicating thermostability. The predicted GRAVY was 0.077, validating the hydrophobic nature of the construct. CoV2-TMEP was found to be non-allergenic by AllergenFP and with a predicted probability of antigenicity of 0.194704 determined by ANTIGENpro.

The immunoinformatic analysis of the CoV2-TMEP construct predicted that the selected epitopes should be able to elicit antigenic responses in the designed sequence. The generated epitope/allele tables used to analyze the population coverage for eliciting T cell responses showed that 91.16% of the world population could be covered by MHC class I epitopes and 91.59% by MHC class II epitopes from CoV2-TMEP, with the lowest coverages in Trinidad and Tobago (0% for MHC-I), Spanish Jewish individuals (0% for MHC-II) and Wales (6.88% for MHC-I and 4.35% for MHC-II) (Appendix A).

To validate the antigenicity predictions of the synthetic CoV2-TMEP protein in preclinical studies, we generated the DNA-CoV2-TMEP vaccine vector that was characterized in vitro by Western blotting and immunofluorescence analyses using a specific antibody against the FLAG tag. As shown in Figure 1B, CoV2-TMEP protein was detected at the expected size (around 35 KDa) in cell extracts from transfected cells at different time-points, but not in the supernatants. CoV2-TMEP forms aggregates that increase over time, with a localization that resembles that of inflammasomes, which are cytosolic multiprotein oligomers of the innate immune system involved in the activation of inflammatory responses (Figure 1C). Thus, we analyzed the potential association of CoV2-TMEP with inflammasome by Western blotting analysis using an antibody against the Nod-like receptor protein 3 (NLRP3). NLRP3 functions as a pattern recognition receptor (PRR) that recognizes pathogen-associated molecular patterns (PAMPs) from bacterial, viral and fungal infections among others and that, together with the adaptor ASC/PYCARD protein and caspase-1, form the NLRP3 inflammasome [30]. As observed in Figure 1D, an induction in the levels of NLRP3 expression was detected in DNA-CoV2-TMEP-transfected cells, indicating that the CoV2-TMEP construct was actively sensed by NLRP3. In addition, since growing evidence suggests that many of the signaling molecules known to regulate programmed cell death can also modulate inflammasome activation in a cell-intrinsic manner [31], we next evaluated the cleavage of poly ADP-ribose polymerase (PARP) in extracts from HEK-293T cells transfected with DNA-ϕ or DNA-CoV2-TMEP vectors. In human PARP, the cleavage separates PARP N-terminal DNA binding domain (24 KDa) from its C-terminal catalytic domain (89 KDa) [32], making it a marker for detecting apoptotic cells. As shown in Figure 1D, the 116 KDa full-length PARP was partially cleaved (89 KDa) in cells transfected with DNA-CoV2-TMEP, while in DNA-ϕ-transfected cells this cleavage was not produced. This result suggests that the presence of CoV2-TMEP construct within the cells is sensed by the innate immune system triggering an apoptotic response. To further confirm this observation, we analyzed the CoV2-TMEP-NLRP3 association by confocal microscopy. As observed in Figure 1E, both proteins colocalized at discrete points in the cytosol of transfected cells, suggesting again that the expression of the multiepitopic CoV2-TMEP protein could enhance inflammasome formation and the subsequent activation of inflammatory and apoptotic responses.

### 3.2. CoV2-TMEP Expression from a DNA Vector Enhances Immune Cell Recruitment in Muscle and DLNs

We next evaluated in vivo the potential impact of the expression of CoV2-TMEP protein from a DNA vector on the recruitment of immune cells to the muscle (inoculation site) and DLNs of C57BL/6JOlaHsd mice vaccinated as depicted in Figure 2A.

In muscle cells (Figure 2B), we observed a continuous increase in the number of B cells (BCs), T cells (TCs) and conventional dendritic cells (cDCs) from day 1 to 7 that could be associated with tissue injury caused by the inoculation process, since no differences were observed between DNA- and PBS-treated groups. In contrast, the number of neutrophils (NOs) progressively decreased from day 1 to 7, suggesting NOs migration from the site of inoculation to secondary lymphoid organs. Interestingly, the local expression of CoV2-TMEP protein induced a significant increase in the number of monocyte-derived cells (moCs) and natural killer (NK) cells. In DLNs (Figure 2C), the inoculation of DNA-CoV2-TMEP induced a significant early inflammatory response at day 1 characterized by the recruitment of BCs, TCs and NKs, and a significant increase in the number of moCs differentiated from recruited monocytes and resident cDCs, probably mediated by the recruitment of cDC-precursor cells (preDCs). DNA-CoV2-TMEP immunization also induced an increase in the number of cDCs that phenotypically resemble tissue-derived DCs that most probably have migrated from the muscle to the lymph node at day 7.

In summary, the multiepitopic CoV2-TMEP protein delivered in vivo by a DNA vector induced an increase in the recruitment of different immune cells with critical roles in innate and adaptive immune responses.

### 3.3. Homologous DNA-CoV2-TMEP Prime/Boost Vaccination Regimen Elicits Robust and Polyfunctional SARS-CoV-2-Specific T Cell Responses in C57BL/6 Mice

To evaluate the immunogenicity of the multiepitopic CoV2-TMEP protein, we characterized the SARS-CoV-2-specific T cell responses elicited in mice by DNA-CoV2-TMEP vector when administered in homologous prime/boost combination following the schedule shown in Figure 3A. Lymphocytes derived from spleen or lung of immunized mice were stimulated ex vivo for 6 h with SARS-CoV-2 S, M, N and E peptide pools and incubated with specific antibodies to identify T cell lineage (CD3, CD4 and CD8) and effector cytokines (IFN-γ, IL-2 and TNF-α) and degranulation (CD107a) to define responding cells. SARS-CoV-2-specific CD4 and CD8 T cells were determined by the percentage of T cells with CD4 or CD8 phenotype that produced IFN-γ and/or IL-2 and/or TNF-α and/or expressed CD107a.

We first evaluated the total number of T cells with CD4 or CD8 phenotype present in the spleen and lung of both immunization groups. As observed in Figure 3B for each stimulus used, the total number of CD4 or CD8 T cells in both organs was significantly increased in DNA-CoV2-TMEP-immunized group compared with the group of mice immunized with the empty vector.

Next, we evaluated the SARS-CoV-2-specific CD4 and CD8 T cell responses (Figure 3C). For CD4 T cells, SARS-CoV-2-specific responses were higher in the spleen than in the lung, with a greater magnitude of CD4^+^ T cells observed in mice immunized with DNA-CoV2-TMEP compared to control group in both organs. CD4^+^ T cell responses were primarily directed against S and N peptide pools. For CD8 T cells, the responses were higher in the lung than in the spleen, with a higher magnitude of CD8^+^ T cells in mice immunized with DNA-CoV2-TMEP compared to control group in both organs. CD8^+^ T cell responses were mainly directed against S followed by M and N peptide pools.

We also evaluated the polyfunctional profile of the SARS-CoV-2-specific T cell responses by the pattern of cytokine and degranulation marker secretion by activated T cells. Based on the analysis of IFN-γ, IL-2, TNF-α and CD107a, eight (for CD4 T cells) or seven (for CD8 T cells) SARS-CoV-2-specific CD4 or CD8 T cell populations were identified (Figure 3D). Vaccine-induced CD4 T cell responses in spleens were polyfunctional, with 50% of CD4^+^ T cells exhibiting two, three or four functions, while in lungs the response was less polyfunctional, with 30% of CD4^+^ T cells exhibiting two to four functions. Vaccine-induced CD8 T cell responses in spleens were highly polyfunctional, with more than 75% of CD8^+^ T cells exhibiting two, three or four functions, while in lungs the response was again less polyfunctional, with 30% of CD8^+^ T cells exhibiting two to four functions.

### 3.4. In Vitro Characterization of an MVA-Based Vector Expressing the Bi-Cistronic CoV2-BMEP-P2A-TMEP Multiepitopic Protein

With the aim of evaluating the effect of the combined expression of both B and T cell activating constructs from the same vector, we generated and characterized an MVA-based recombinant virus expressing the bi-cistronic CoV2-BMEP-P2A-TMEP multiepitopic protein targeting functional and conserved regions of SARS-CoV-2 structural proteins (MVA-CoV2-B2AT, Figure 4A).

The insertion of the *CoV2-BMEP-P2A-TMEP* gene into the viral genome and the purity of the MVA-CoV2-B2AT viral preparations were analyzed by PCR using primers annealing in the flanking regions of TK locus. As shown in Figure 4B, the *CoV2-BMEP-P2A-TMEP* gene was successfully inserted into the TK locus of MVA genome, evidenced by the detection of the expected 2594 bp product, while no contamination with parental MVA-WT (873 bp product) was observed in MVA-CoV2-B2AT P2 and P3 viral stocks. These results were further confirmed by DNA sequencing.

The expression and processing of the bi-cistronic CoV2-BMEP-P2A-TMEP protein from the MVA vector over time was assessed by Western blotting. As shown in Figure 4C, a gradual increase in the levels of CoV2-BMEP (35 KDa) and CoV2-TMEP (34 KDa) proteins, at their expected sizes, was detected over time, starting at 16 hpi. Additionally, a higher molecular weight product was also detected, which may correspond to the full-length non-processed CoV2-BMEP-P2A-TMEP protein. Through immunofluorescence microscopy using anti-N or anti-HA specific antibodies, the CoV2-BMEP-P2A-TMEP protein was mainly detected forming aggregates in the cytoplasm of infected cells (Figure 4D).

Finally, the *CoV2-BMEP-P2A-TMEP* gene was stably integrated and maintained within the viral genome of the recombinant MVA virus as revealed the Western blotting analysis of 24 individual plaques picked up after seven successive infection passages in DF-1 cells. As shown in Figure 4E, 24 out of 24 plaques (100% stability) expressed both proteins, highlighting the genetic stability of the MVA-CoV2-B2AT vector.

### 3.5. Effect of CoV2-BMEP-P2A-TMEP Expression from an MVA Vector on Immune Cell Recruitment in Muscle and DLNs

We next analyzed the impact of the expression of the bi-cistronic CoV2-BMEP-P2A-TMEP multiepitopic protein from MVA-CoV2-B2AT vector on the recruitment of immune cells to the muscle and DLNs of C57BL/6JOlaHsd mice vaccinated following the schedule depicted in Figure 5A.

Similar numbers of innate immune cell populations were detected in muscle cells (Figure 5B) and DLNs (Figure 5C) of MVA-CoV2-B2AT- and MVA-WT-immunized mice, except for Ly6C^low^MHCII^high^moCs and NOs in the muscle, where a significant increase in both populations was observed in the MVA-WT-immunized group compared to MVA-CoV2-B2AT-vaccinated group. This difference could be attributed to the inherent pro-inflammatory profile of the MVA vector.

### 3.6. Homologous MVA-CoV2-B2AT Prime/Boost Vaccination Regimen Effectively Induces Neutralizing Antibodies and Reduces SARS-CoV-2 Beta Variant Replication in K18-hACE2 Tg Mice

We previously reported that MVA-CoV2-BMEP-vaccinated animals with high titers of NAbs were fully protected against mortality after SARS-CoV-2 challenge (MAD6 isolate) [7]. To evaluate the protective potential of MVA-CoV2-B2AT compared to MVA-CoV2-BMEP against SARS-CoV-2 Beta variant infection, we conducted an efficacy study in susceptible K18-hACE2 Tg mice following the homologous MVA/MVA vaccination schedule depicted in Figure 6A.

#### 3.6.1. Analysis of the SARS-CoV-2-Specific Humoral Response at Pre-Challenge 

We determined NAbs against SARS-CoV-2 virus (MAD6 and Beta, Delta and Omicron BA.1 VoCs) induced by MVA-CoV2-BMEP or MVA-CoV2-B2AT vectors at 26 days post-prime (d26) and 25 days post-boost (d52). As shown in Figure 6B, a single dose of either MVA-CoV2-BMEP or MVA-CoV2-B2AT (d26) induced SARS-CoV-2-specific NAbs against the MAD6 strain and the Beta, Delta and Omicron BA.1 VoCs. These levels were further enhanced after the second dose (d52), reaching NT50 titers of 10^3^–10^4^. A slight trend towards lower NAbs induction in the MVA-CoV2-B2AT group compared to the MVA-CoV2-BMEP group was observed. In summary, we can conclude that the combined expression of synthetic CoV2-BMEP and CoV2-TMEP maintains a similar NAbs-inducing capacity as CoV2-BMEP alone.

#### 3.6.2. Restricted SARS-CoV-2 Virus Replication Induced by Vaccination 

The analysis of SARS-CoV-2 virus replication in lung samples at day 5 post-challenge revealed a reduction in viral replication in both MVA-vaccinated groups compared to unvaccinated challenged control group, as indicated by lower levels of genomic *RdRp* and subgenomic *E* mRNAs. This reduction was significant for subgenomic *E* mRNA (Figure 6C). This observation correlated with the analysis of viral yields in lung and nasal turbinate samples (Figure 6D), where MVA-vaccinated mice showed a reduction in infectious virus compared to challenged control group. These findings suggest that the combined expression of CoV2-BMEP and CoV2-TMEP proteins from the bi-cistronic MVA-CoV2-B2AT vector induces the suppression of viral replication and a reduction in viral yields comparable to that achieved by MVA-CoV2-BMEP alone.

### 3.7. MVA-CoV2-B2AT Combined with ISG15 Adjuvant Enhanced Magnitude, Breadth and Polyfunctional Profile of the SARS-CoV-2-Specific T Cell Response in C57BL/6 Mice

Finally, we decided to characterize the SARS-CoV-2-specific T cell responses elicited in mice by MVA-CoV2-B2AT vector when administered in homologous prime/boost regimen alone or in combination with an MVA vector expressing either the wild-type or the mutated form of murine ISG15 as adjuvant (the immune advantage of ISG15 for HIV-1 antigens has been previously reported [8]) in order to improve the immunogenicity of the bi-cistronic MVA-based vector. For this, four groups of C57BL/6JOlaHsd mice were immunized by the i.m. route following the immunization schedule depicted in Figure 7A. The analysis of the SARS-CoV-2-specific CD4 and CD8 T cells was performed as described above.

As shown in Figure 7B, the SARS-CoV-2-specific T cell response was polarized towards the CD8 T cell compartment in all immunization groups. The strongest CD4 T cell response was detected in mice immunized with the combination MVA-CoV2-B2AT and MVA-Δ3-ISG15AA (group 3), with the response evenly distributed across S, M and N peptide pools. Similarly, the highest CD8 T cell response was also detected in group 3, followed by group 2 (MVA-CoV2-B2AT + MVA-Δ3-ISG15GG) and group 1 (MVA-CoV2-B2AT + MVA-Δ3-GFP). The CD8^+^ T cell response was mainly directed against the S peptide pool.

Based on the analysis of IFN-γ, IL-2, TNF-α and CD107a, six SARS-CoV-2-specific CD8 T cell populations were identified. As observed in Figure 7C, vaccine-induced CD8 T cell responses were highly polyfunctional in all immunization groups, except in group 1 (MVA-CoV2-B2AT + MVA-Δ3-GFP), with 75% of CD8^+^ T cells exhibiting two, three or four functions in group 2 (MVA-CoV2-B2AT + MVA-Δ3-ISG15GG) and 50% in group 3 (MVA-CoV2-B2AT + MVA-Δ3-ISG15AA). CD8 T cells simultaneously producing CD107a + IFN-γ + TNF-α followed by IL-2 alone, CD107a alone and CD107a + IFN-γ + IL-2 + TNF-α were the most representative populations.

In summary, these results indicate that the expression of the bi-cistronic CoV2-BMEP-P2A-TMEP multiepitopic protein from an MVA vector is able to elicit SARS-CoV-2-specific T cell responses, which may contribute, together with the induction of NAbs, to the reduction in virus replication and viral yields observed in the efficacy study conducted in K18-hACE2 Tg mice. Furthermore, the co-expression of ISG15 further improves the specific immune response by increasing its magnitude, broadening its scope and enhancing its polyfunctional profile.

## 4. Discussion

The ongoing COVID-19 pandemic has highlighted the need for vaccines and therapies capable of addressing the evolving challenges posed by SARS-CoV-2, particularly the emergence of variants that compromise vaccine efficacy. While current vaccines, primarily targeting the S protein, have successfully mitigated the impact of the pandemic, their protection is limited by factors such as the rapid decay of NAbs, poor neutralization of emerging VoCs, inadequate induction of mucosal immunity and focus on the S protein alone, overlooking other viral proteins that could broaden and strengthen the effectiveness and durability of the immune response [33,34,35].

To address these challenges, next-generation vaccines aim to provide broad and durable protection by targeting conserved viral elements beyond the S protein [36]. To drive innovation in vaccine design, the careful selection of appropriate antigens is crucial for ensuring broad and cross-reactive immunity. Computational approaches are being used to identify antigens that maximize sequence conservation across different coronaviruses while maintaining their immunogenicity [37,38]. In this context, multiepitopic vaccine candidates offer a promising strategy to enhance both the immunogenicity and long-term efficacy.

In a previous study, we designed the CoV2-BMEP multiepitopic construct including B cell epitopes from conserved regions of SARS-CoV-2 S, M and N structural proteins that, when expressed by an MVA vector, induced strong B and T cell responses in C57BL/6 mice and provided full protection against SARS-CoV-2 mortality in K18-hACE2 Tg mice with high titers of NAbs [7]. To extend these findings, and based on evidence highlighting the critical role of T cells in controlling COVID-19 [39,40,41,42,43], particularly given their resistance to viral mutations and potential to provide long-lasting protection against emerging variants even in the absence of neutralizing antibodies [44,45,46], we detail here the design, characterization and preclinical evaluation of two novel polyvalent multiepitopic vaccine candidates: CoV2-TMEP and CoV2-BMEP-P2A-TMEP. These constructs, which include conserved and functional B and T cell epitopes from SARS-CoV-2 S, M, N and E structural proteins, were efficiently expressed from a DNA vector (DNA-CoV2-TMEP) or an MVA vector (MVA-CoV2-B2AT) in cultured cells.

The HLA coverage of the 12 patches included in the TMEP multiepitopic protein was determined using in silico computational algorithms, which have been shown to achieve over 85%-90% accuracy in predicting epitope-HLA binding. However, while these predictions are reliable, further experimental validation is needed to confirm their immunogenicity. In our study, we selected conserved T cell regions from the SARS-CoV-2 structural S, M, N and E proteins based on experimentally validated immunodominant epitopes described in previous studies of vaccinated and convalescent COVID-19 patients across diverse populations. For example, epitopes such as N105-113 (SPRWYFYYL), restricted by HLA-B07:02, have shown strong CD8^+^ T cell responses in over 75% of HLA-B07:02-positive individuals, as confirmed in multiple studies, including Grifoni et al. [47,48]. Similarly, the N322-330 (MEVTPSGTWL) epitope, restricted by HLA-B*40:01, was shown to induce robust T cell responses in up to 88.9% of convalescent patients. These epitopes are not only immunodominant but have also shown cross-reactivity with new variants of SARS-CoV-2, reinforcing their importance for long-term, global population coverage.

The expression of CoV2-TMEP protein from the DNA vector in mice immunized by i.m. route resulted in a significant enhancement of the recruitment of immune cells, including moCs and NKs, in the muscle and DLNs at early time-points. This recruitment is essential for the initiation and amplification of the immune response. The induction of inflammasome formation and apoptotic responses observed in DNA-CoV2-TMEP-transfected cells further underscores the potential of this construct to effectively activate innate immune pathways in vivo. Indeed, in SARS-CoV-2 infection, the magnitude of the initial viral load [49] and the efficacy of the innate immune response, particularly that mediated by type I interferons (IFN) [50,51], are crucial for establishing the subsequent adaptive response and determining the clinical outcome. The critical role of effective IFN signaling in acute infection has been clearly demonstrated [52,53].

Despite the critical role of innate responses in controlling viral infections [54], coordinated cellular immunity is also essential for effective disease control [55]. In our study, we observed that a homologous DNA-CoV2-TMEP prime/boost vaccination regimen significantly increased SARS-CoV-2-specific CD4 and CD8 T cell responses in both the spleen and lungs. These responses targeted multiple viral proteins (S and N for CD4; and S, followed by M and N for CD8) and were polyfunctional, indicating the potential of DNA-CoV2-TMEP to induce robust and sustained immunity. These findings align with previous research on various T cell-based multiepitopic constructs expressed from DNA as vaccine candidates against SARS-CoV-2, which have shown promising results. For instance, a T cell vaccine candidate composed entirely of in silico-predicted epitopes from the entire SARS-CoV-2 genome was evaluated in vivo and successfully induced SARS-CoV-2-specific T cells against 15 out of the 17 included epitopes in immunized C57BL/6J mice, protecting K18-hACE2-susceptible mice against lethal disease even in the absence of NAbs [56]. Similarly, two T cell-based pan-sarbecovirus multiantigen DNA vaccines encoding a large set of antigens shared across sarbecoviruses derived from non-structural proteins nsp7, nsp8, nsp12 and nsp13 (CoVAX_ORF1ab) or from structural proteins M, N and S (CoVAX_MNS) induced both CD8^+^ and CD4^+^ T cell responses to shared epitopes in C57BL/6 mice and provided partial protection against a lethal SARS-CoV-2 challenge in K18-hACE2-susceptible mice [57]. These studies underscore the potential of T cell-based strategies in providing effective and broad protection against SARS-CoV-2 and related coronaviruses.

The robust T cell responses observed with the DNA-CoV2-TMEP vector align with key parameters that have been associated with protection in SARS-CoV-2 infection. The early development of cytotoxic CD8^+^ T cell, which has been linked to effective viral clearance [58] and mild disease outcomes [59], is particularly noteworthy. The induction of strong Th1 and CD8^+^ T cell responses in asymptomatic or mild cases further underscores the importance of a well-coordinated cellular immune response [48,60,61]. In contrast, severe COVID-19 cases often exhibit strong B cell responses but limited T cell responses, highlighting the critical role of T cells in mitigating disease severity [59,62]. Moreover, higher numbers of tissue-resident T cells in the lungs have been associated with protection from severe COVID-19 [63,64]. Given that the lungs are the primary site of COVID-19 infection, the induction of lung-specific CD8^+^ T cells following systemic homologous DNA-CoV2-TMEP vaccination may be critical for protection. The functional capacity of these T cells is also a key determinant of clinical outcome, with CD8^+^ T cells expressing high levels of effector molecules being associated with improved outcomes in acute COVID-19 [65]. Furthermore, N protein-specific T cell responses have been associated with less severe disease and lower viral loads in animal studies [66]. In humans, the N protein elicits some of the strongest CD8^+^ T cell responses during natural infection, with polyfunctional N-specific CD8 T cells being associated with mild disease [67]. Supporting the role of the N protein as an effective antigenic target, CD8^+^ T cell responses against the NP105–113 epitope restricted by HLA-B*07:02 have been shown to induce strong anti-viral immunity and correlate with protection from severe disease [68]. Overall, these findings suggest that targeting both B and T cell epitopes, particularly those from the N protein, could be crucial for broad and effective protection in future COVID-19 vaccines [69].

Accumulated evidence from numerous longitudinal COVID-19 patient cohort studies have shown that a coordinated adaptive immune response, characterized by the rapid expansion of both SARS-CoV-2-specific CD4 and CD8 T cells and virus-specific NAbs, within seven days of infection, is correlated with protection from severe disease [41,55,70,71,72]. Building on these findings, in this study we also explore the combined expression of B and T cell targeting immunogens through the design of the polyfunctional bi-cistronic CoV2-BMEP-P2A-TMEP multiepitopic protein expressed from an MVA vector. The homologous MVA-CoV2-B2AT prime/boost vaccination regimen elicited NAbs against various SARS-CoV-2 VoCs and reduced viral replication and viral yields against the Beta variant in susceptible K18-hACE2 mice, demonstrating that the co-expression of both CoV2-BMEP and CoV2-TMEP proteins does not compromise the induction of NAbs or the protective efficacy compared to CoV2-BMEP alone. Furthermore, the inclusion of a mutated ISG15 variant (ISG15AA) as an adjuvant significantly enhanced the magnitude, breadth and polyfunctional profile of the SARS-CoV-2-specific immune response, especially for the CD8 T cell subset. These results align with our previous findings on the importance of ISG15 as an immune adjuvant in HIV-1 vaccine development in which the expression of mutated ISG15AA (unable to perform ISGylation) from an MVA vector in combination with the HIV-1 vaccine candidate MVA-B (expressing the HIV-1 Env and Gag-Pol-Nef antigens from clade B) induced a higher IFN-I production compared to the wild-type ISG15 (ISG15GG), indicating a more robust innate immune response that correlated with an increase in the magnitude and polyfunctional profile of the HIV-1-specific CD8 T cells [8]. The adjuvant effect of ISG15 observed in the context of SARS-CoV-2 infection could also be related with higher levels of IFN-I production, but further studies should be conducted to elucidate the exact molecular mechanisms through which ISG15 enhances the immune response.

In support of this strategy, various studies have shown that bi-cistronic constructs, like our BMEP-P2A-TMEP, can induce robust B and T cell responses, which are crucial for effective protection against SARS-CoV-2 and its variants. For example, scPR8-RBD-M2, a single-cycle influenza virus-based SARS-CoV-2 vaccine encoding 2A peptide-based bi-cistronic protein cassette of the SARS-CoV-2 receptor-binding domain (RBD) and influenza matrix 2 (M2) protein, elicited robust mucosal and systemic humoral immune responses and cell-mediated immunity to influenza virus NP and SARS-CoV-2 S proteins in mice, with serum samples exhibiting neutralizing activity against pseudotyped viruses carrying SARS-CoV-2 S from various variants, albeit with varying potency [73]. Another study with pGO-1002, a bi-cistronic synthetic DNA vaccine encoding consensus sequences of SARS-CoV-2 S and ORF3a antigens, elicited specific humoral and cellular responses to both antigens, effectively preventing viral replication in different animal models [74]. This vaccine was also evaluated in a phase I clinical trial, inducing SARS-CoV-2 S-specific BAbs in 95.5% of participants, NAbs in 55.5% of participants and T cell responses in 97.8% of participants, remaining high through 48 weeks [75]. Similarly, a chimpanzee adenoviral vector-based vaccine expressing a modified full-length S (vS) and conserved T cell epitopes from ORF1, ORF3 and M (TCEs) separated by P2A linker (AdC68-vST-vtRBM) elicited higher NAbs responses to all variants, as well as lgA, lgG, GC B cells, long-lived plasma cells, tissue-resident memory T cells and systemic memory T cells, which conferred full protection against BA.2 infection in hACE2 Tg mice [76]. Despite some concerns regarding the co-delivery of genes in bi-cistronic vectors [77], our findings and other recent studies highlight the viability of such approaches for inducing comprehensive immune responses.

Overall, our study highlights the potential of both polyvalent CoV2-TMEP and CoV2-BMEP-P2A-TMEP multiepitopic proteins, expressed from DNA or MVA vectors, as next-generation SARS-CoV-2 vaccines. Their ability to induce broad, robust and protective immune responses, along with extensive population coverage, positions them as promising candidates to overcome the limitations of current vaccines. The enhancement of the magnitude, breadth and polyfunctional profile of the specific immune response by ISG15 further underscores the role of adjuvants in vaccine development. Future research should focus on optimizing these constructs, evaluating their efficacy and assessing their protective effects against emerging VoCs. This evaluation should include an analysis of the long-term immune response, potentially enhanced by the inclusion of CoV2-TMEP construct, coadministration with other vaccine platforms (e.g., mRNA) and preclinical testing in additional animal models (e.g., ferrets or non-human primates) to obtain more translatable results, overcoming the limitations of the murine model. Such research will be essential to fully understand the potential of these vaccine candidates within the evolving SARS-CoV-2 vaccine landscape.

## Figures and Tables

**Figure 1 vaccines-12-01213-f001:**
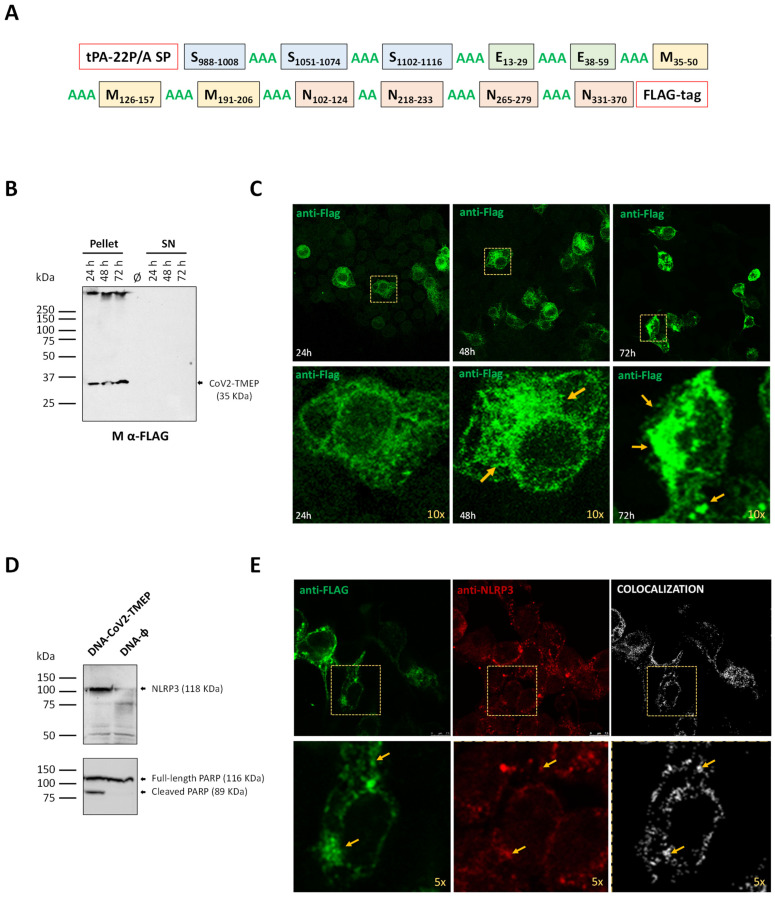
In vitro characterization of a DNA vector expressing the multiepitopic CoV2-TMEP protein. (**A**) Schematic representation of the chimeric CoV2-TMEP multiepitopic vaccine targeting T cells. (**B**) Time-course expression of CoV2-TMEP protein. HEK-293T cells were transfected as detailed in the Materials and Methods section and CoV2-TMEP expression in pellets and supernatants (SN) was analyzed by Western blotting using the mouse monoclonal anti-FLAG antibody. (**C**) Subcellular localization of CoV2-TMEP protein. HeLa cells were transfected and processed for immunofluorescence microscopy as indicated in the Materials and Methods section. Green staining: mouse monoclonal anti-FLAG antibody followed by anti-mouse Alexa Fluor 488 antibody. Yellow arrows: CoV2-TMEP forming aggregates in magnified images (10×). (**D**) Analysis of NLRP3 induction and PARP cleavage. HEK-293T cells were transfected as described in the Materials and Methods section and the expression of NLRP3 (118 KDa; upper panel) and PARP (full-length: 116 KDa; cleaved: 89 KDa; lower panel) was analyzed by Western blotting using specific antibodies. (**E**) Subcellular colocalization of CoV2-TMEP protein with inflammasome. HeLa cells were transfected and processed for confocal microscopy as detailed in the Materials and Methods section. A mouse monoclonal anti-FLAG antibody and a rabbit polyclonal anti-NLRP3/NALP3 antibody followed by the corresponding secondary antibodies conjugated with Alexa Fluor 488 (green staining; FLAG detection) or Alexa Fluor 594 (red staining; NLRP3 detection) were used. Colocalization signal is indicated in white. Yellow arrows: Discrete colocalization signals in magnified images (5×). One representative optical section from three independent experiments is shown.

**Figure 2 vaccines-12-01213-f002:**
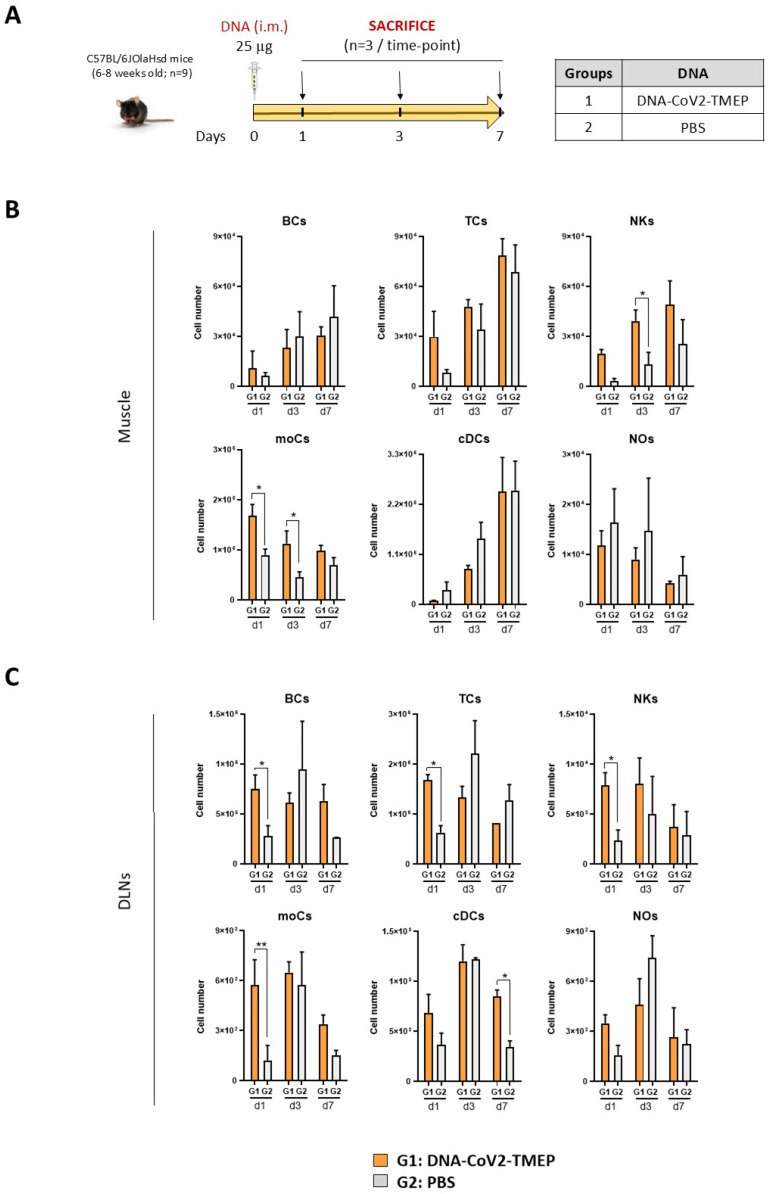
Innate immune response elicited in muscle and DLNs from mice immunized with a DNA vector expressing the polyvalent multiepitopic CoV2-TMEP protein. (**A**) Immunization schedule. Mice were inoculated with DNA-CoV2-TMEP or PBS by intramuscular (i.m.) route. At the indicated days post-inoculation, total muscle from the site of inoculation and DLNs were harvested and processed as described in the Materials and Methods section. (**B**,**C**) Immune cell populations present in muscle (**B**) and DLNs (**C**) determined by flow cytometry. Data are shown as mean and SD. BCs: B cells; TCs: T cells; NKs: natural killer cells; moCs: monocyte-derived cells; cDCs: conventional dendritic cells; NOs: neutrophils. *, *p* < 0.05; **, *p* < 0.005.

**Figure 3 vaccines-12-01213-f003:**
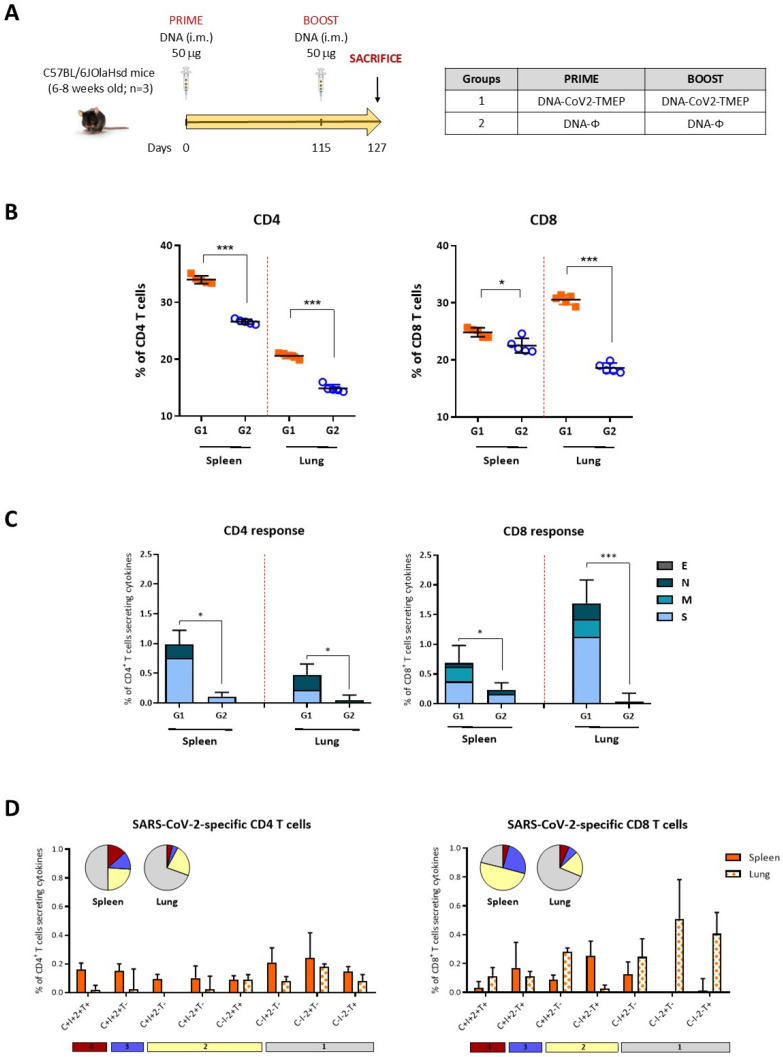
SARS-CoV-2-specific T cell adaptive immune responses elicited by DNA-CoV2-TMEP in C57BL/6 mice when administered in homologous regimen. (**A**) Immunization schedule. Mice were inoculated with DNA-CoV2-TMEP or DNA-ϕ by i.m. route at days 0 and 115. At 12 days post-boost, animals were sacrificed, and spleens and lungs were harvested and processed for ICS assay, as described in the Materials and Methods section. (**B**) Magnitude of the total CD4 (left panel) or CD8 (right panel) T cells in the spleens and lungs of immunized mice. Each colored form represents the value obtained for each stimulus (S, M, N, E and RPMI). The 95% confidence interval (CI) is represented. (**C**) Magnitude of the total SARS-CoV-2-specific CD4 (left) or CD8 (right) T cells at 12 days post-boost after the stimulation of lymphocytes derived from spleen or lung with SARS-CoV-2 peptide pools. The total value of each group represents the sum of the percentages of SARS-CoV-2-specific CD4 or CD8 T cells expressing CD107a and/or producing IFN-γ and/or IL-2 and/or TNF-α against SARS-CoV-2 peptide pools. Data are background-subtracted. The 95% CI is represented. (**D**) Polyfunctional profile of the SARS-CoV-2-specific CD4 (left) or CD8 (right) T cells in DNA-CoV2-TEMP-immunized mice in spleen and lung. The positive combinations of the responses are indicated on the *x* axis, while the percentages of the functionally different cell populations within the total CD4 or CD8 T cells are represented on the *y* axis. Specific responses are grouped and color-coded based on the number of functions. All data are background-subtracted. The 95% CI is shown. C: CD107a; I: IFN-γ; 2: IL-2; T: TNF-α. *, *p* < 0.05; ***, *p* < 0.001.

**Figure 4 vaccines-12-01213-f004:**
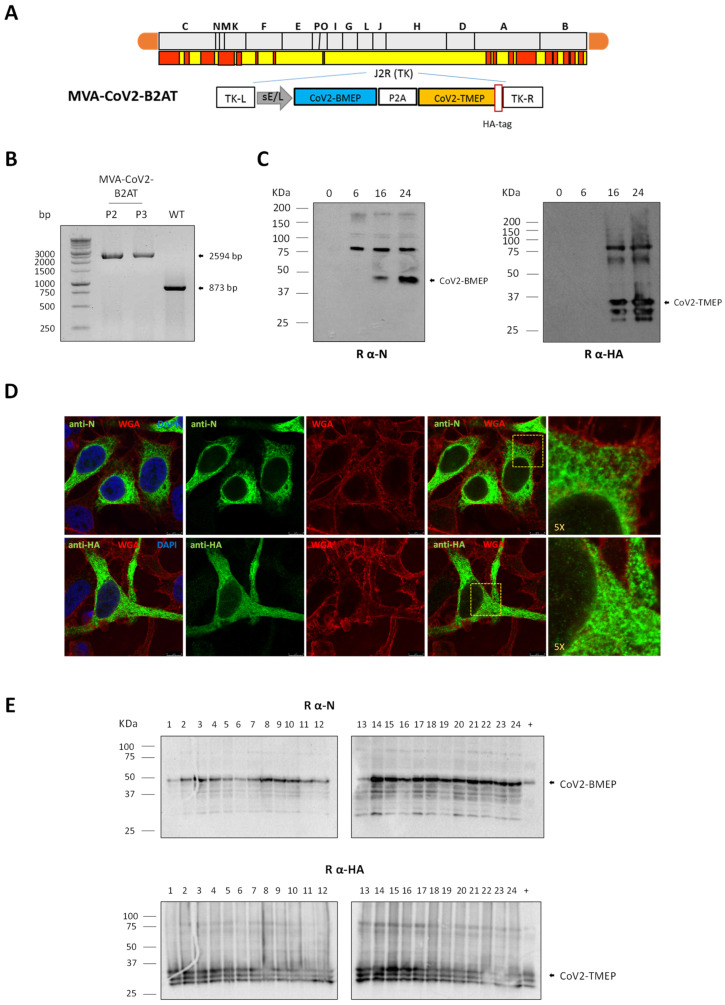
In vitro characterization of an MVA-based recombinant virus expressing the multiepitopic CoV2-BMEP-P2A-TMEP protein. (**A**) Scheme of the thymidine kinase (TK) locus of MVA-CoV2-B2AT recombinant virus. (**B**) Confirmation of *CoV2-BMEP-P2A-TMEP* gene insertion. DNA extracted from DF-1 cells infected with MVA-WT or MVA-CoV2-B2AT (P2 and P3 viral preparations) viruses as described in the Materials and Methods section was used as template for the analysis of the TK locus by PCR using primers TK-L and TK-R spanning TK flanking regions. (**C**) Time-course expression of CoV2-BMEP-P2A-TMEP protein. DF-1 cells infected with MVA-CoV2-B2AT as detailed in the Materials and Methods section were harvested at the indicated times post-infection and analyzed by Western blotting using a rabbit polyclonal anti-SARS-CoV-2 N antibody (left; CoV2-BMEP) or a rabbit polyclonal anti-SARS-CoV-2 HA antibody (right; CoV2-TMEP). (**D**) Subcellular localization of CoV2-BMEP-P2A-TMEP protein. HeLa cells infected with MVA-CoV2-B2AT for 16 h were harvested and processed for immunofluorescence microscopy as described in the Materials and Methods section. Red staining: WGA-Alexa Fluor 555; green staining: rabbit polyclonal anti-N antibody (upper panels) or rabbit polyclonal anti-HA antibody (lower panels) followed by the corresponding secondary antibody conjugated with Alexa Fluor 488; blue staining: DAPI. (**E**) Stability of the CoV2-BMEP-P2A-TMEP protein. The expression of CoV2-BMEP-P2A-TMEP protein in 24 individual plaques from MVA-CoV2-B2AT stability passage 7 was analyzed by Western blotting using a rabbit polyclonal anti-N antibody (upper) or a rabbit polyclonal anti-HA antibody (lower). Lane (+): MVA-CoV2-B2AT P2 stock.

**Figure 5 vaccines-12-01213-f005:**
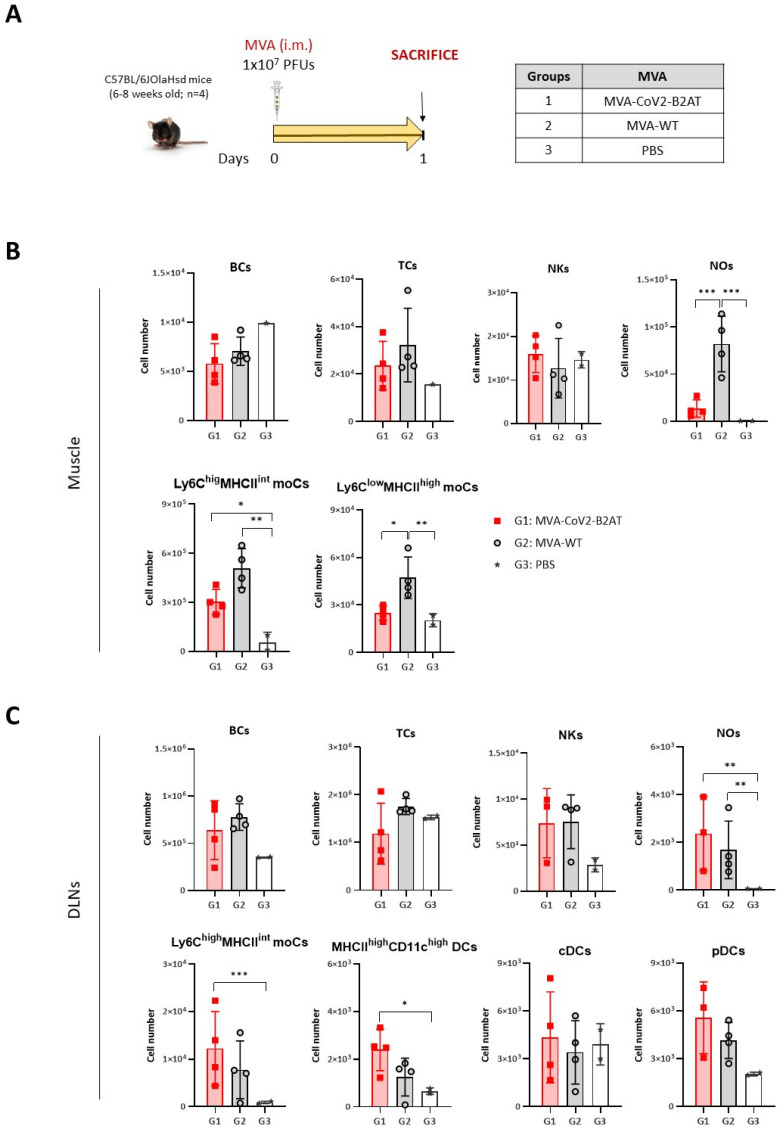
Innate immune response elicited in muscle and DLNs from mice immunized with an MVA vector expressing the multiepitopic CoV2-BMEP-P2A-TMEP protein. (**A**) Immunization schedule. Mice were immunized with MVA-WT or MVA-CoV2-B2AT or PBS-treated by i.m. route. At day 1 post-inoculation, total muscle from the site of inoculation and DLNs were excised and processed as described in the Materials and Methods section. (**B**,**C**) Immune cell populations present in muscle (**B**) and DLNs (**C**) determined by flow cytometry. Data are shown as colored forms for each animal with mean and SD. BCs: B cells; TCs: T cells; NKs: natural killer cells; NOs: neutrophils; moCs: monocyte-derived cells; DCs: dendritic cells; cDCs: conventional dendritic cells; pDCs: plasmacytoid dendritic cells. *, *p* < 0.05; **, *p* < 0.005; ***, *p* < 0.001.

**Figure 6 vaccines-12-01213-f006:**
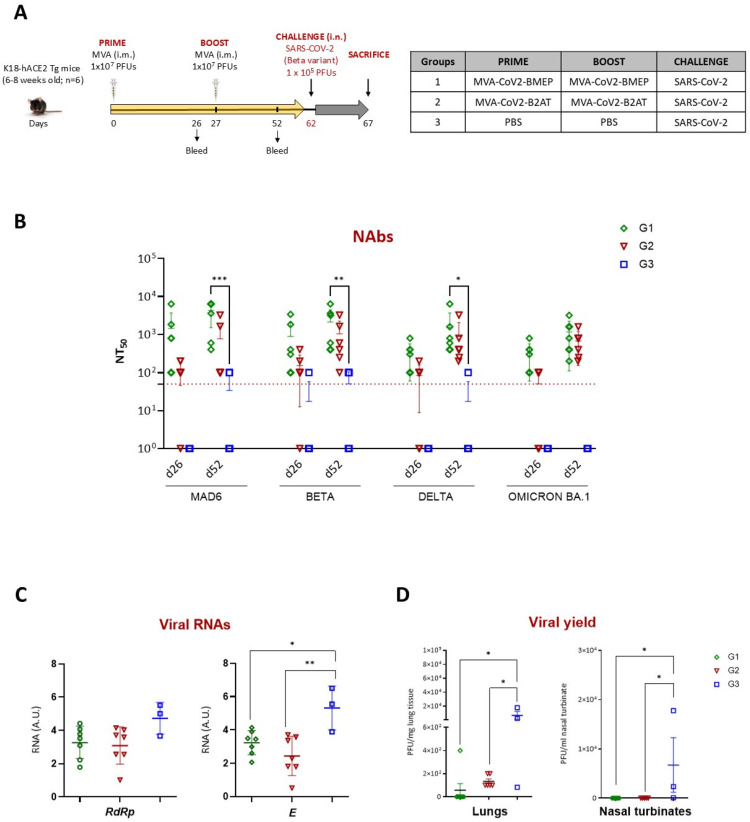
Efficacy study in K18-hACE2 Tg mice. (**A**) Immunization schedule. K18-hACE2 Tg mice were immunized by i.m. route with the indicated viruses or with PBS at days 0 and 27 and challenged 5 weeks later (d62) by intranasal (i.n.) route with SARS-CoV-2 virus (Beta variant). At 5 days post-challenge, mice were sacrificed, and lung, nasal turbinate and serum samples were harvested and processed as described in the Materials and Methods section. (**B**) SARS-CoV-2 NAbs (NT_50_) in serum from immunized mice against SARS-CoV-2 MAD6 and Beta, Delta and Omicron BA.1 VoCs at d26 and d52 determined by MNT assay. Data are shown as forms for each animal with mean and SD. Red dashed line: lower limit of detection (LLD) of the assay. (**C**) Virus replication in lung samples. Genomic (*RdRp*) and subgenomic (*E*) SARS-CoV-2 RNAs were detected by RT-qPCR post-challenge. RNA levels [in arbitrary units (A.U.)] for each animal are represented as forms with mean and SD from duplicates; relative values are referred to as a naïve lung sample. (**D**) SARS-CoV-2 infectious virus in lungs (left) and nasal turbinates (right). Data are shown as forms for each animal with mean (PFUs/mg of lung tissue or PFUs/mL of nasal turbinate tissue) and SD from duplicates. *, *p* < 0.05; **, *p* < 0.005; ***, *p* < 0.001.

**Figure 7 vaccines-12-01213-f007:**
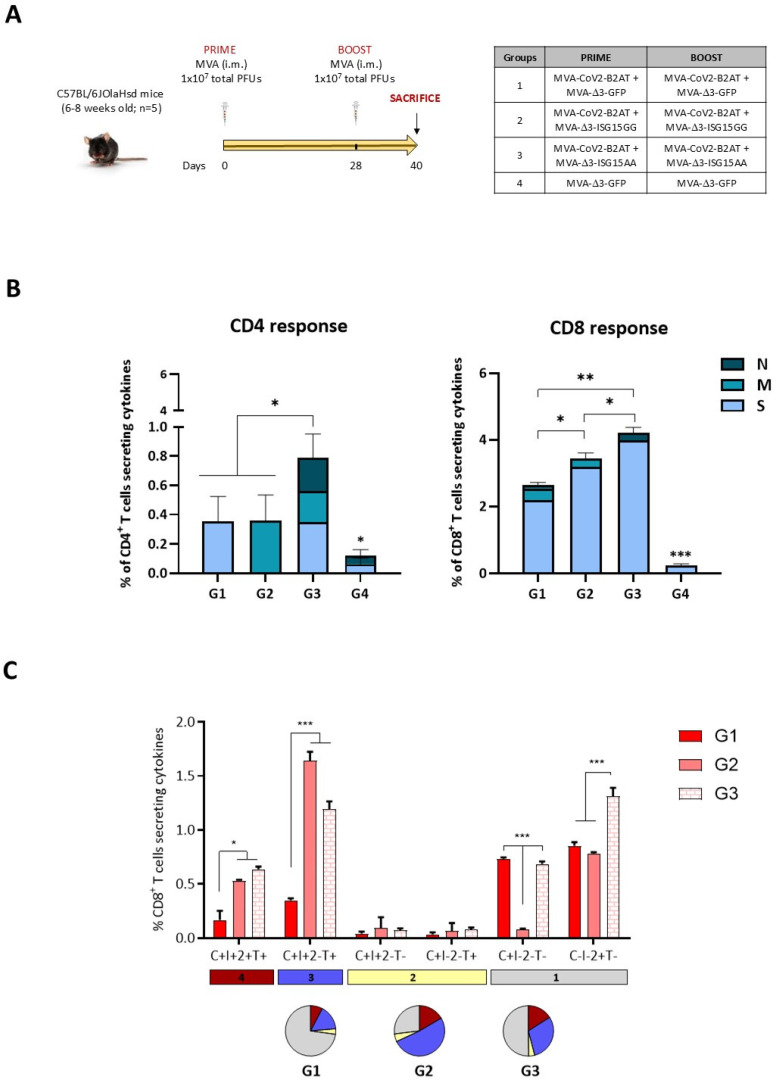
SARS-CoV-2-specific T cell adaptive immune responses elicited by MVA-CoV2-B2AT in C57BL/6 mice when administered in homologous regimen in the presence or absence of ISG15 as adjuvant. (**A**) Immunization schedule. Mice were inoculated with the indicated vector combinations by i.m. route at days 0 and 28. At 12 days post-boost, animals were sacrificed and spleens were harvested and processed for ICS assay, as described in the Materials and Methods section. (**B**) Magnitude of the total SARS-CoV-2-specific CD4 (left) or CD8 (right) T cells at 12 days post-boost after the stimulation of splenocytes with SARS-CoV-2 peptide pools. The total value of each group represents the sum of the percentages of SARS-CoV-2-specific CD4 or CD8 T cells expressing CD107a and/or producing IFN-γ and/or IL-2 and/or TNF-α against SARS-CoV-2 peptide pools. Data were background-subtracted. The 95% CI is represented. (**C**) Polyfunctional profile of the SARS-CoV-2-specific CD8 T cells in immunized mice. The positive combinations of the responses are indicated on the *x* axis, while the percentages of the functionally different cell populations within the total CD8 T cells are represented on the *y* axis. Specific responses are grouped and color-coded based on the number of functions. All data were background-subtracted. The 95% CI is shown. C: CD107a; I: IFN-γ; 2: IL-2; T: TNF-α. *, *p* < 0.05; **, *p* < 0.005; ***, *p* < 0.001.

**Table 1 vaccines-12-01213-t001:** Epitope sequences included in the CoV2-TMEP synthetic protein.

Peptide Sequence	Viral Protein	Start	End
EAEVQIDRLITGRLQSLQTYV	S	988	1008
SFPQSAPHGVVFLHVTYVPAQEKN	S	1051	1074
WFVTQRNFYEPQIIT	S	1102	1116
IVNSVLLFLAFVVFLLV	E	13	29
RLCAYCCNIVNVSLVKPSFYVY	E	38	59
LQFAYANRNRFLYIIK	M	35	50
GTILTRPLLESELVIGAVILRGHLRIAGHHLG	M	126	157
SGFAAYSRYRIGNYKL	M	191	206
KDLSPRWYFYYLGTGPEAGLPYG	N	102	124
ALALLLLDRLNQLESK	N	218	233
TKAYNVTQAFGRRGP	N	265	279
LTYTGAIKLDDKDPNFKDQVILLNKHIDAYKTFPPTEPKK	N	331	370

## Data Availability

The raw data supporting the conclusions of this article will be made available by the authors without undue reservation.

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
