# Peer review of "B and T Cell Bi-Cistronic Multiepitopic Vaccine Induces Broad Immunogenicity and Provides Protection Against SARS-CoV-2"

_vaccines, 2024, doi:10.3390/vaccines12111213_

Round 1
Reviewer 1 Report
Comments and Suggestions for Authors
The manuscript "B and T cell bi-cistronic multiepitopic vaccine induces broad immunogenicity and provides protection against SARS-CoV-2", which explores two vaccine candidates based on multi-cistronic epitopes aimed at inducing both B and T cell immune responses against SARS-CoV-2. The manuscript addresses a critical need in the development of second-generation vaccines by aiming for a more robust and long-lasting immunity against SARS-CoV-2 variants. It presents convincing data on the efficacy of the vaccine candidates in murine models, demonstrating their ability to induce robust immune responses in C57BL/6 and K18-hACE2 mice. Additionally, the inclusion of an adjuvant based on a mutated form of ISG15 is both novel and promising.
I find the choice of conserved viral proteins (S, M, N, and E) to be a smart approach for inducing both broad-spectrum neutralizing antibodies and T cell responses, potentially improving coverage of emerging variants.
Major Comments
- While the manuscript mentions the use of the mutated ISG15 adjuvant to enhance immune responses, it would be useful to delve deeper into the exact molecular mechanisms through which ISG15 enhances the immune response. Additionally, a more thorough comparison with other adjuvants used in similar studies could help evaluate its relative effectiveness.
- The study includes concerning SARS-CoV-2 variants, such as Beta and Omicron BA.1, but a more detailed discussion on how these variants specifically affect the efficacy of the proposed vaccines would be valuable. Since neutralization capacity varies between variants, providing more details about the magnitude of the reduction in neutralization for each variant, alongside potential correlations between cellular and humoral immunity, would be beneficial.
- The authors mention statistical significance analyses, but the methodology for selecting cut-off points (e.g., for fold change or immune response significance) could be better explained.
- Although strong induction of T cell responses and neutralizing antibodies is observed in the short term, it remains unclear how long these responses persist. A longer-term follow-up to assess the durability of the immune response is recommended.
- It is not specified whether studies were conducted to determine the optimal dosage of MVA- and DNA-based vaccines to maximize response without increasing toxicity or adverse effects. Future studies should explore this to refine vaccine administration.
- Currently, the study uses K18-hACE2 mice, which express the human ACE2 receptor, limiting the extrapolation to humans. Testing additional models (e.g., ferrets or non-human primates) that could provide more translatable results is recommended. If not possible, discussing this limitation is crucial.
- Since multiepitopic vaccines can be combined with other platforms, I suggest conducting co-administration experiments with other vector or mRNA vaccines to evaluate potential synergistic effects or interferences in immune response.
- A detailed study of toxicity in both murine models and other models is essential to evaluate the long-term safety of the proposed formulations, especially considering the use of ISG15 as an adjuvant.
- Several paragraphs in the manuscript contain material that appears to be directly taken from other articles, particularly in the Materials and Methods section. For example, Section 2.2 appears to be derived from a previous publication by the authors (DOI: 10.3389/fimmu.2023.1160065, PMID: 37404819; PMCID: PMC10316789). In this case, it would be important for the authors to appropriately reference their earlier work when describing methods that have been previously published, especially if it gives the impression that the methods or results are being presented as novel when they have been reported before. Moreover, the article could benefit from a more original description of the methods, possibly by adapting them or emphasizing specific adjustments made for the current study, thus clearly distinguishing the work from previous publications.
Author Response
Dear reviewer 1,
Thank you very much for taking the time to review this manuscript. Please see the attached file to find the detailed responses to your comments and the corresponding corrections in track changes in the re-submitted version of the manuscript.
Best regards,
Beatriz Perdiguero.

Reviewer 2 Report
Comments and Suggestions for Authors The authors developed two vaccine candidates: DNA-CoV2-TMEP, expressing the multiepitopic protein containing immunodominant and conserved T cell regions from SARS-CoV-2 structural proteins; and MVA-CoV2-B2AT, encoding a bi-cistronic multiepitopic construct that combines conserved B and T cell overlapping regions from SARS-CoV-2 structural proteins. Both candidates were assessed in vitro and in vivo and demonstrated their ability to induce robust immune responses as well as protection against SARS-CoV-2. The experiments are well designed and results are sounded to support the conclusion. One concern is that how the 12 peptides containing T-cell epitopes can cover more than 90% global population. The HLA restriction of 12 peptides only was defined only by in silico computational algorithms. The authors should provide the real-world experimental data to confirm the persistent and dominant epitopes in infected cohorts, especially confirm the population coverage.
Author Response
Dear reviewer 2,
Thank you very much for taking the time to review this manuscript. Please see the attached file to find the detailed responses to your comments and the corresponding corrections in track changes in the re-submitted version of the manuscript.
Best regards,
Beatriz Perdiguero.

Reviewer 3 Report
Comments and Suggestions for Authors
Previously, the authors reported the effectiveness of B cell multiepitope protein (BMEP) as a vaccine candidate against SARS-CoV-2. In this manuscript, the authors expanded their research scope and evaluated T cell multiepitope protein (TMEP) for a vaccine candidate with or without BMEP against SARS-CoV-2. Both proteins elicited T-cell responses against SARS-CoV-2. In a mouse model, the BMEP-TMEP fusion protein blocked viral replication at a similar efficacy to that of BMEP alone. Based on these results, the authors concluded that these multiepitope proteins are promising candidates for the next-generation COVID-19 vaccine.
Overall, I do not have any significant concerns about the technical aspects of this manuscript. However, I do have concerns about the authors’ conclusions. Although the authors empathized with the importance of T cell response in fighting against SARS-CoV-2 and concluded that T cell multiepitope proteins could be a promising candidate for the next generation of the COVID-19 vaccine, the results show otherwise. As Fig. 6 demonstrates, the anti-viral efficacy of BMEP-TMEP fusion protein is, at best, as good as or lower than BMEP. If that is the case, there is no reason to use TMEP for vaccine development. It might be prudent to tone down the vaccine potential of TMEP in the Abstract and Discussion.
Other comments
Line 472: The half-life was determined to be..
: The half-life is just a predicted one, not experimentally measured. Therefore, the above should be changed to “The half-life was estimated to be..” or something similar.
Line 479: The immunoinformatic analysis of CoV2-TEMP construct confirmed..
: Again, it was estimated but not confirmed. I think no in-silico tools can confirm anything. They estimate or predict.
Lines 505, 509: phi is missing in “DNA- ”
Lines 576, 577: alpha is missing in “TNF-“
Fig. 4C: The intact BMEP-P2A-TMEP fusion protein was detected with anti-N-terminal antibody at 6 h. However, it was not detected with an anti-HA tag antibody. I wonder why.
Fig. 4E: It seems that the unprocessed protein was not detected. Why?
Fig. 6D: The viral yield in the lungs seems higher with G2 as compared with G1. Is it statistically significant?
Line 815: enhancement of immune cells -> enhancement of the recruitment of immune cells (?)
Comments on the Quality of English LanguageOverall, the writing is OK, although there are some parts that is difficult to follow.
Author Response
Dear reviewer 3,
Thank you very much for taking the time to review this manuscript. Please see the attached file to find the detailed responses to your comments and the corresponding corrections in track changes in the re-submitted version of the manuscript.
Best regards,
Beatriz Perdiguero.

Round 2
Reviewer 1 Report
Comments and Suggestions for Authors
The authors have satisfactorily addressed my concerns. I believe the work can be published. However, the authors still need to find a way to ensure that the cited methodologies are not verbatim. Again, that would be my only significant observation
Author Response
Dear Reviewer 1,
We are grateful for your positive assessment of the response to your comments.
Regarding the verbatim of some parts of the M&M section, it is a difficult task to modify it since the different techniques reported in the manuscript are widely used in this type of research and have been optimized in previous works from our lab. Nevertheless, we have tried as far as possible to modify the wording to try to reduce this verbatim in the modified re-submitted version.
We hope that your concern will be now satisfied.
Best regards,
Beatriz Perdiguero.
Reviewer 3 Report
Comments and Suggestions for Authors
This is a revised manuscript evaluating the effectiveness of a T cell epitope-based COVID-19 vaccine. Through revision, the authors have addressed and explained my previous comments, and I no longer have any concerns regarding the manuscript.
Author Response
Dear Reviewer 3,
We would like to thank you for all your suggestions and comments which have contributed significantly to improving the quality of the manuscript.
Best regards,
Beatriz Perdiguero.